# DualFast: Dual-Speedup Framework for Fast Sampling of Diffusion Models

## Abstract

Diffusion probabilistic models (DPMs) have achieved impressive success in visual generation. While, they suffer from slow inference speed due to iterative sampling. Employing fewer sampling steps is an intuitive solution, but this will also introduces discretization error. Existing fast samplers make inspiring efforts to reduce discretization error through the adoption of high-order solvers, potentially reaching a plateau in terms of optimization. This raises the question: can the sampling process be accelerated further? In this paper, we re-examine the nature of sampling errors, discerning that they comprise two distinct elements: the widely recognized discretization error and the less explored approximation error. Our research elucidates the dynamics between these errors and the step by implementing a dual-error disentanglement strategy. Building on these foundations, we introduce an unified and training-free acceleration framework, DualFast, designed to enhance the speed of DPM sampling by concurrently accounting for both error types, thereby minimizing the total sampling error. DualFast is seamlessly compatible with existing samplers and significantly boost their sampling quality and speed, particularly in extremely few sampling steps. We substantiate the effectiveness of our framework through comprehensive experiments, spanning both unconditional and conditional sampling domains, across both pixel-space and latent-space DPMs.

## 1 Introduction

Diffusion probabilistic models (DPMs) Sohl-Dickstein et al. (2015); Ho et al. (2020); Song et al. (2020b) have demonstrated impressive success across a broad spectrum of tasks, including image synthesis Dhariwal & Nichol (2021); Rombach et al. (2022); Ramesh et al. (2022); Saharia et al. (2022), image editing Meng et al. (2021), video generation Ho et al. (2022); Blattmann et al. (2023), voice synthesis Chen et al. (2020), etc. Compared with other generative models such as GANs Goodfellow et al. (2014) and VAEs Kingma & Welling (2013), DPMs not only exhibit superior sample quality but also benefit from a more robust training methodology and an advanced guided sampling technique. However, the inference of DPMs usually requires multiple model evaluations (NFEs), which hinders their practical deployment.

Recently, there has been a surge in endeavors to expedite the sampling processes of DPMs Salimans & Ho (2022); Meng et al. (2023); Song et al. (2023; 2020a); Liu et al. (2022); Lu et al. (2022a); Zhang & Chen (2022); Yu et al. (2023); Lu et al. (2022b); Zhao et al. (2023); Zheng et al. (2023), which are broadly categorized into training-based model distillation and training-free fast sampling approaches. Distillation-based techniques, notable for facilitating generation in a minimal number of steps, are somewhat curtailed by a complex distillation procedure and the necessity for model-specific distillation, which constrain their broader adoption. Conversely, fast samplers Song et al. (2020a); Zhang et al. (2022); Liu et al. (2022); Lu et al. (2022a); Zhang & Chen (2022); Lu et al. (2022b); Zhao et al. (2023); Wang et al. (2023); Zhang et al. (2023); Xu et al. (2023); Zheng et al. (2023) enjoy wider popularity for their inherent training-free quality, allowing seamless integration with readily available pre-trained DPMs. These methods predominantly leverage probability flow ordinary differential equations (ODEs) and can be formulated in a general continuous exponential integrator form. Their main strategy centers on minimizing discretization errors that emerge from approximating such continuous integration with large step size discretization (small sampling steps), and potentially reaching a plateau in terms of optimization. However, an intriguing question arises:

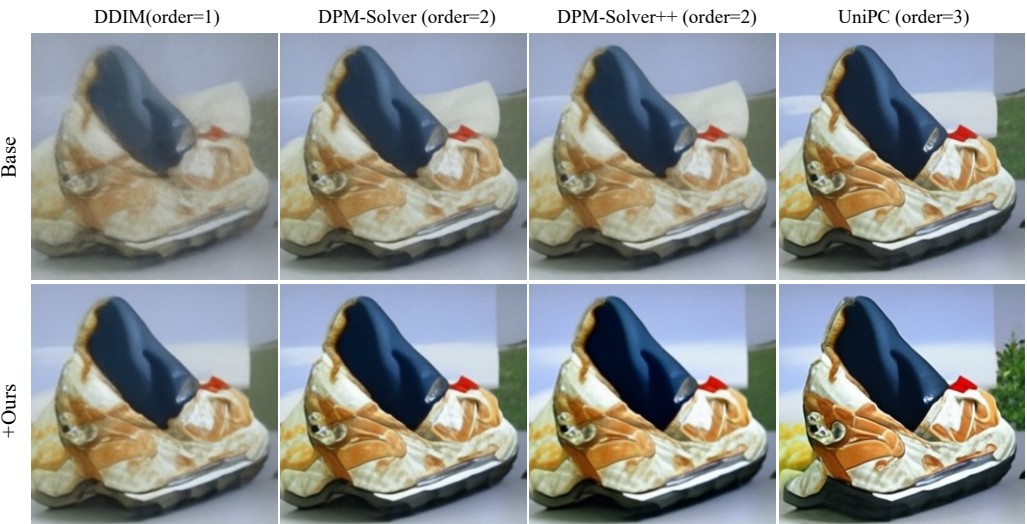

Figure 1: Qualitative comparisons between existing solvers and our method with class condition. All images are generated by sampling from ADM Dhariwal & Nichol (2021) trained on ImageNet 256×256 with only 7 number of function evaluations (NFEs). We show that our proposed method can significantly elevate the sample quality with better details and contrast than previous base samplers.

*besides minimizing the discretizaton error, is it possible to further elevate the sampling speed and quality with fewer-step inference?*

In this paper, we start with a detailed examination of the constituents of the total sampling error, discerning it into discretization and approximation errors. The former part originates from the discretization of the continuous exponential integrator, whereas the latter part emerges due to the neural network's imprecise estimation of the ground truth vector field (score function). Prevailing fast solvers predominantly address the discretization error, inadvertently ignoring the approximation error, thus leaving space for enhanced inference acceleration. Further, we introduce a dual-error disentanglement strategy aimed at disentangle these two sub-errors from the total sampling error, subsequently elucidating their interrelations and associations with the step. Our findings indicate that both errors are of comparable magnitude, underscoring the criticality and significance of reducing the approximation error for further sampling acceleration. Moreover, we discover that the approximation error monotonically decreases as step $t$ increases, a relationship that is pivotal for guiding the reduction of approximation error in later design.

With these insights, we propose an unified and training-free acceleration framework, DualFast, for the fast sampling of DPMs by taking both discretization and approximation errors into consideration to further reduce the total sampling error. To mitigate the discretization error, DualFast readily incorporates existing fast ODE solver techniques. For the approximation error, we innovatively design the critical approximation error reduction strategy, crafted to integrate fluidly with current fast ODE solvers, thereby precipitating an additional acceleration. Furthermore, we elucidate mathematically the process of integrating this approximation error reduction strategy into several representative fast solvers, including DDIM Song et al. (2020a), DPM-Solver Lu et al. (2022a), and DPM-Solver++ Lu et al. (2022b), which consist of various orders and prediction modes.

The efficacy of our DualFast framework is rigorously substantiated through comprehensive experiments that span a diverse range of models, datasets, solvers, and sampling steps. Specifically, these experiments are structured to encompass two sampling types (unconditional and conditional generation), two condition types (class and text conditions), two sampling spaces (pixel-space and latent-space DPMs), samplers of different orders (1-order DDIM, 2-order DPM-Solver and DPM-Solver++), different prediction modes (noise and data predictions), and various sampling steps. DualFast significantly improves the sampling quality and efficiency over previous solvers on all the conducted experiments, especially with extremely limited sampling steps. Qualitative comparisons, as depicted in Figure 1, reveal a consistent advantage of our method in producing images of superior structure, details, and color contrast compared to those generated by the base solvers.

## 2 RELATED WORK

### 2.1 DIFFUSION PROBABILISTIC MODELS

Diffusion probabilistic models (DPMs) Sohl-Dickstein et al. (2015); Ho et al. (2020); Song et al. (2020b) transform complex data distribution into simple noise distribution and learn to recover data from noise. The forward diffusion process starts from clean data sample $x_0$ and repeatedly injects Gaussian noise to a simple normal distribution $x_T \sim \mathcal{N}(\mathbf{0}, \sigma^2 \mathbf{I})$ at time T > 0 for some $\sigma > 0$. The corresponding transition kernel $q_{t|0}(x_t \mid x_0)$ is as follows:

$$q_{t|0}(x_t \mid x_0) = \mathcal{N}(x_t | \alpha_t x_0, \sigma_t^2 \mathbf{I}), \tag{1}$$

where the *signal-to-noise-ratio* (SNR) equals $\alpha_t^2 / \sigma_t^2$.

**Training process.** Diffusion models are trained by optimizing a variational lower bound (VLB). For each step $t$, the denoising score matching loss is the distance between two Gaussian distributions, written as:

$$\min_\theta \mathbb{E}_{x_0, \epsilon, t} \left[ \omega(t) \| \epsilon_\theta(x_t, t) - \epsilon \|_2^2 \right], \tag{2}$$

where $\omega(t)$ is weighting function, $\epsilon \sim q(\epsilon) = \mathcal{N}(\epsilon | \mathbf{0}, \mathbf{I})$, and $x_t = \alpha_t x_0 + \sigma_t \epsilon$.

**ODE-based sampling process.** Sampling from DPMs can be achieved by solving the following diffusion ODEs Song et al. (2020b):

$$\frac{\mathrm{d}x_t}{\mathrm{d}t} = f(t) x_t + \frac{g^2(t)}{2\sigma_t} \epsilon_\theta(x_t, t), \quad x_T \sim \mathcal{N}(\mathbf{0}, \sigma^2 \mathbf{I}), \tag{3}$$

where the coefficients $f(t) = \frac{\mathrm{d} \log \alpha_t}{\mathrm{d}t}$, $g^2(t) = \frac{\mathrm{d}\sigma_t^2}{\mathrm{d}t} - 2 \frac{\mathrm{d} \log \alpha_t}{\mathrm{d}t} \sigma_t^2$.

### 2.2 FAST SAMPLING OF DPMS

A large step size in stochastic differential equations (SDEs) violates the randomness of the Wiener process Kloeden & Platen (1992) and often causes non-convergence. Certain methods Guo et al. (2023); Gonzalez et al. (2024) propose to accelerate SDE solvers but still lag behind ODE solvers in speed [1]. Restart sampling Xu et al. (2023) combines the SDE and ODE to boost sampling quality, and also mentions the concept of the approximation error, but it neglects to present the reason, disentanglement, and impact of this error. Chen et al. Chen et al. (2023) explored to decompose the score function of the linear subspace data, but under the low-dimensional linear subspace assumption. For faster sampling, *probability flow ODE* (Song et al., 2020b) is usually considered as a better choice. Recent works Lu et al. (2022a); Zhang & Chen (2022); Lu et al. (2022b) find that ODE solvers built on exponential integrators Hochbruck & Ostermann (2010) appear to have faster convergence than directly solving the diffusion ODEs. Given an initial value $x_s$ at time $s > 0$, the solution $x_t$ at each time $t < s$ of diffusion ODEs can be analytically computed as Lu et al. (2022a):

$$x_t = \frac{\alpha_t}{\alpha_s} x_s - \alpha_t \int_{\lambda_s}^{\lambda_t} e^{-\lambda} \epsilon_\theta(\hat{x}_\lambda, \lambda) \mathrm{d}\lambda, \tag{4}$$

where the ODE is changed from the time (t) domain to the log-SNR ($\lambda$) domain by the change-of-variables formula, and the notation $\hat{x}_\lambda$ denote change-of-variables. Based on the exponential integrator, existing ODE solvers approximate $\epsilon_\theta(\hat{x}_\lambda, \lambda)$ via Taylor expansion at time $\lambda_s$.

$$x_t = \frac{\alpha_t}{\alpha_s} x_s - \alpha_t \sum_{n=0}^{k-1} \epsilon_\theta^{(n)}(\hat{x}_{\lambda_s}, \lambda_s) \int_{\lambda_s}^{\lambda_t} e^{-\lambda} \frac{(\lambda - \lambda_s)^n}{n!} \mathrm{d}\lambda + \mathcal{O}(h_t^{k+1}), \tag{5}$$

where $k$ denotes the order of the Taylor expansion, also known as the order of the solver, and $h_t = \lambda_t - \lambda_s$ is the step size. Obviously, high order solver reduces the discretization error to $\mathcal{O}(h_t^{k+1})$. Further, it can be simplified into a general form as follows:

$$x_t = \frac{\alpha_t}{\alpha_s} x_s - \sigma_t(e^{h_t} - 1) D_t. \tag{6}$$

---

[1] We omit the comparison with SDE-based samplers in this paper due to their randomness and slow speed.

Existing ODE solvers only differ in $\boldsymbol{D}_t$. For the first-order DDIM solver, $\boldsymbol{D}_t = \boldsymbol{\epsilon}_\theta(\boldsymbol{x}_s, s)$. DPM-Solver++ Lu et al. (2022b) considers rewriting equation 4 using $\boldsymbol{x}_\theta$ instead of $\boldsymbol{\epsilon}_\theta$. UniPC Zhao et al. (2023) proposes a predictor-corrector method to refine the prediction. A common thread in these approaches is that they attempt to reduce discretization error part via high order Taylor approximation in equation 5, while ignore the approximation error induced in equation 4 when replacing the true score function with the network approximation $\boldsymbol{\epsilon}_\theta(\hat{\boldsymbol{x}}_\lambda, \lambda)$.

## 3 METHOD

### 3.1 MOTIVATION AND INSIGHTS

In order to identify the errors in the inference stage, we delve into the transition from the exact error-free solution of diffusion ODE to its practical implementation form in equation 6 as follows:

$$
\begin{aligned}
\boldsymbol{x}_t &= \frac{\alpha_t}{\alpha_s}\boldsymbol{x}_s - \alpha_t \int_{\lambda_s}^{\lambda_t} e^{-\lambda} \left[ -\sigma_t \nabla_{\boldsymbol{x}} \log q_t(\boldsymbol{x}_t) \right] \mathrm{d}\lambda \quad \text{\textit{(exact solution)}} \\
&\approx \frac{\alpha_t}{\alpha_s}\boldsymbol{x}_s - \alpha_t \int_{\lambda_s}^{\lambda_t} e^{-\lambda} \boldsymbol{\epsilon}_\theta(\hat{\boldsymbol{x}}_\lambda, \lambda) \mathrm{d}\lambda \quad \text{\textit{(approximation error induced)}} \\
&\approx \frac{\alpha_t}{\alpha_s}\boldsymbol{x}_s - \sigma_t(e^{h_t} - 1)\boldsymbol{D}_t. \quad \text{\textit{(discretization error induced)}}
\end{aligned}
\tag{7}
$$

Here, $\boldsymbol{D}_t$ signifies the polynomial of $\boldsymbol{\epsilon}_\theta(\hat{\boldsymbol{x}}_\lambda, \lambda)$, mirroring varying degrees of discrete approximations to the continuous integral. The above formulation reveals that the induced error consists of two parts: discretization error and approximation error. The approximation error arises from the substitution of the precise score function $-\sigma_t \nabla_{\boldsymbol{x}} \log q_t(\boldsymbol{x}_t)$ with the network's approximation $\boldsymbol{\epsilon}_\theta(\hat{\boldsymbol{x}}_\lambda, \lambda)$ due to the denoising network's imprecise score function estimation. Subsequently, discretization error emerges when simulating the continuous integral through discrete implementation. While previous fast samplers predominantly aim at diminishing discretization error via higher-order Taylor expansions, introducing variously ordered samplers such as 1-order DDIM and 2-order DPM-Solver, they overlook the criticality of approximation error. It is noticeable that while some prior methods Xu et al. (2023); Hunter et al. (2023) may recognize the presence of approximation error, they conclude the exploration at this point. In stark contrast, this derivation is the start of our study. We conduct comprehensive analyses and studies to thoroughly unlock the approximation error, and proceed to integrating this error into existing solvers for further acceleration with an unified and general ODE-based acceleration framework.

### 3.2 DUAL-ERROR DISENTANGLEMENT

With the above insights on the total sampling error, we then delve into dissecting the properties of these two errors. To achieve this, we introduce a dual-error disentanglement strategy, effectively isolating these sub-errors as depicted in Figure 2. Specifically, within the time interval $[s, t]$, we construct three distinct transition processes, each subjected to varying levels of sampling error. Initially, we define the exact data distributions $\mathcal{P}(\boldsymbol{x}_s)$ and $\mathcal{P}(\boldsymbol{x}_t)$, which are free of both approximation and discretization errors. These distributions are derived from the pristine image distribution $\mathcal{P}(\boldsymbol{x}_0)$, employing the transition kernel outlined in equation 1. This procedure establishes an optimal, error-free transition from distribution $\mathcal{P}(\boldsymbol{x}_s)$ to $\mathcal{P}(\boldsymbol{x}_t)$. Our next objective is to formulate a second distribution transition from $\mathcal{P}(\boldsymbol{x}_s)$ to $\mathcal{P}(\boldsymbol{x}_t^s)$, which is solely afflicted by approximation error, exempt from discretization error. To this end, we adopt extremely small step size to minimize the discretization error when approximating the continuous integral. Subsequently, we chart the third distribution transition from $\mathcal{P}(\boldsymbol{x}_s)$ to $\mathcal{P}(\boldsymbol{x}_t^l)$, suffering from both discretization and approximation errors, facilitated by a larger step size.

Originating from the identical data distribution $\mathcal{P}(\boldsymbol{x}_s)$, we now derive three distributions $\mathcal{P}(\boldsymbol{x}_t)$, $\mathcal{P}(\boldsymbol{x}_t^s)$, and $\mathcal{P}(\boldsymbol{x}_t^l)$, thereby disentangling approximation and discretization errors within the time range $[s, t]$. Notably, the discrepancy between $\mathcal{P}(\boldsymbol{x}_t)$ and $\mathcal{P}(\boldsymbol{x}_t^s)$ illuminates the approximation error, whereas the divergence between $\mathcal{P}(\boldsymbol{x}_t^s)$ and $\mathcal{P}(\boldsymbol{x}_t^l)$ manifests the discretization error. Note that though it is infeasible to clearly separate these two errors or acquire their precise error values, the above disentanglement analysis provides the rough error magnitude level and changing trend. An

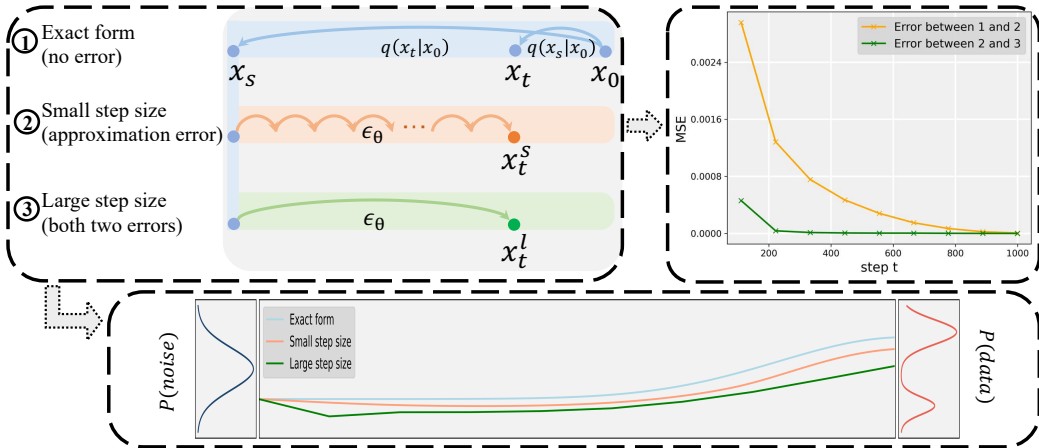

Figure 2: (Left): Dual-Error Disentanglement. For the identical time period, we map out three distinctive distribution transition processes, each subjected to a unique combination of errors. The top one represents exact data distribution and is free of errors. The middle one is liberated from discretization error owing to an extremely fine-grained step size. The bottom one, with a coarse step size, succumbs to both errors. (Right): The MSE curve along the step $t$. The error between operation 1 and 2 manifests approximation error, and discretization error can be reflected between operation 2 and 3. We divide the total $T = 1000$ step into 9 time periods, with each of length 111. For every time period, we adopt NFE=111 and NFE=1 to get $x_t^s$ and $x_t^l$, respectively. (Bottom): Overview of ODE-based generation process mapping noise to data distribution, where smaller errors correspond to higher probability density region.

illustrative MSE curve of these errors across the step $t$ is depicted in Figure 2. Analysis of the MSE curve yields two pivotal insights: (1) Both errors are of comparable magnitude and the approximation error surpasses the discretization error at most timesteps, highlighting its significant influence in the sampling process and underscoring the criticality of reducing the approximation error for accelerating sampling. (2) The discretization error decreases as the step $t$ increases. This conclusion is also consistent with EDM Karras et al. (2022), which finds that the discretization error is smaller at larger noise level. (3) The approximation error exhibits a strict decline as the step $t$ increases, a principle that subsequently instructs the design of our approximation error reduction strategy. Note that the rationality of employing MSE to measure the distribution discrepancy in diffusion models is illustrated in the supplementary material.

Besides the above findings about the changing trend of approximation error, we make one further step to analyze its intrinsic reason. During training stage, the denoising network is trained to approach the ground-truth Gaussian noise hidden in the input noisy image. With higher noise level in the input $x_t$, the noise pattern is more recognisable and the network also tends to produces smaller MSE error. Accordingly, the network prediction and score function are more accurate as step $t$ gets larger. The similar training loss curve observations in Yu et al. (2023) also support our analyses.

## 3.3 ACCELERATION FRAMEWORK-DUALFAST

Based on the above findings, we introduce our unified and training-free DualFast framework, crafted to synergize with prevailing fast samplers while concurrently mitigating the approximation error for augmented acceleration. For the discretization error, we seamlessly incorporate the Taylor approximation mechanism employed by preceding fast ODE solvers. For the approximation error, we formulate a reduction strategy by leveraging the property of the approximation error that it monotonically decreases with advancing step $t$. Specifically, at inference stage, the input of the denoising network at step $T$ is pure Gaussian noise, and the network will also output the same noise pattern as the input. This means that this input Gaussian noise highly resembles the optimal output. However, as step $t$ gets smaller, the noise level becomes lower, and the original noise pattern is harder to identify from the input. This characteristic, where the network's estimation is more desired at larger step, guides us to substitute the noise prediction at the current step $t$ with that of a preceding, larger step $\tau$.

Now, we take the basic 1-order Taylor approximation sampler as example and show how to get the general from applicable for all existing samplers. As is known, DDIM is the 1-order ODE sampler Lu et al. (2022a). The sampling formulation of DDIM is as follows:

$$\boldsymbol{x}_{t-1}^{base} = \alpha_{t-1}\boldsymbol{x}_\theta(\boldsymbol{x}_t, t) + \sigma_{t-1}\boldsymbol{\epsilon}_\theta(\boldsymbol{x}_t, t). \tag{8}$$

Further, we can rewrite the above equation in the unified form of equation 6 with following $\boldsymbol{D}_{t-1}$:

$$\boldsymbol{D}_{t-1}^{base} = \boldsymbol{\epsilon}_\theta(\boldsymbol{x}_t, t). \tag{9}$$

To reduce the approximation error, we replace the noise estimation part $\boldsymbol{\epsilon}_\theta(\boldsymbol{x}_t, t)$ in equation 9 with $\boldsymbol{\epsilon}_\theta(\boldsymbol{x}_\tau, \tau)$, where $\tau$ is larger than $t$.

$$\boldsymbol{x}_{t-1}^{ours} = \alpha_{t-1}\boldsymbol{x}_\theta(\boldsymbol{x}_t, t) + \sigma_{t-1}\boldsymbol{\epsilon}_\theta(\boldsymbol{x}_\tau, \tau). \tag{10}$$

Similarly, we derive the corresponding $D_t^{ours}$ of equation 11, with full derivation process available in the supplementary material:

$$\boldsymbol{D}_{t-1}^{ours} = (1 + c)\boldsymbol{\epsilon}_\theta(\boldsymbol{x}_t, t) - c\boldsymbol{\epsilon}_\theta(\boldsymbol{x}_{\tau,\tau}) \tag{11}$$

where $c = \frac{1}{e^{h_t} - 1}$ is the mixing coefficient. Given that 1-order approximation is the foundation of the Taylor approximation and the discrepency between equation 9 and equation 9, we now can derive the general form for approximation reduction as follows.

$$\boldsymbol{\epsilon}_\theta^{new}(\boldsymbol{x}_t, t) = (1 + c)\boldsymbol{\epsilon}_\theta(\boldsymbol{x}_t, t) - c\boldsymbol{\epsilon}_\theta(\boldsymbol{x}_\tau, \tau), \tag{12}$$

where $c$ is the mixing coefficient, and step $\tau$ is larger than current step $t$. The mixing coefficient $c$ should monotonically increase as current step $t$ decreases, echoing the tendency that smaller step corresponds to larger approximation error. Besides, as analysed above, we adopt $\tau = T$. The detailed examination and ablation study about the configurations of $c$ and $\tau$ are presented in Sec. 4.3. Besides, while equation 12 is expressed within the context of noise prediction, it is equally applicable to data prediction due to the equivalent transformation between these modes. Moreover, DualFast only needs one NFE per step.

Besides the 1-order DDIM sampler, DualFast also can be seamlessly integrated into existing other fast ODE solvers to achieve further speedup. We provide a thorough mathematical integration process of this approximation error reduction strategy into existing solvers. For instance, we select two more common and representative fast ODE samplers: DPM-Solver, and DPM-Solver++, spanning various orders and prediction modes.

**Ours-DPM-Solver** We apply multi-step, thresholding strategy Lu et al. (2022b) and second-order to DPM-Solver, and get the base version termed as DPM-Solver(2M). Note that the only difference between DPM-Solver(2M) and DPM-Solver++(2M) is the prediction mode.

DPM-Solver reveals that diffusion ODEs have a semi-linear structure and derives the formulation of the solutions by analytically computing the linear part of the solutions, avoiding the corresponding discretization error. Concretely, DPM-Solver(2M) can be directly written in the formation of equation 6 with the corresponding $\boldsymbol{D}_{t-1}$:

$$\boldsymbol{D}_{t-1}^{base} = \boldsymbol{\epsilon}_\theta(\boldsymbol{x}_t, t) + a_1\left[\boldsymbol{\epsilon}_\theta(\boldsymbol{x}_t, t) - \boldsymbol{\epsilon}_\theta(\boldsymbol{x}_{t+1}, t+1)\right], \tag{13}$$

where $a_1$ is the coefficient for the second part of DPM-Solver. Then, we adopt a similar way as in previous DDIM part to reduce the approximation error.

$$\boldsymbol{D}_{t-1}^{ours} = \left[(1 + c)\boldsymbol{\epsilon}_\theta(\boldsymbol{x}_t, t) - c\boldsymbol{\epsilon}_\theta(\boldsymbol{x}_\tau, \tau)\right] + a_1\left[\boldsymbol{\epsilon}_\theta(\boldsymbol{x}_t, t) - \boldsymbol{\epsilon}_\theta(\boldsymbol{x}_{t+1}, t+1)\right]. \tag{14}$$

**Ours-DPM-Solver++** DPM-Solver++ is the sota and default samplers in stable diffusion model. It finds that previous high-order fast samplers suffer from instability issue, and solves the diffusion ODE with the data prediction model. Due to employing different prediction mode, DPM-Solver++ reformulates the implementation equation 6 as follows:

$$\boldsymbol{x}_{t-1} = \frac{\sigma_{t-1}}{\sigma_t}\boldsymbol{x}_t - \alpha_{t-1}(e^{-h_t} - 1)\boldsymbol{D}_{t-1}, \tag{15}$$

where $\boldsymbol{D}_{t-1}$ is expressed in data-prediction $\boldsymbol{x}_\theta(\boldsymbol{x}_t, t)$ as follows:

$$\boldsymbol{D}_{t-1}^{base} = \boldsymbol{x}_\theta(\boldsymbol{x}_t, t) + a_2\left[\boldsymbol{x}_\theta(\boldsymbol{x}_t, t) - \boldsymbol{x}_\theta(\boldsymbol{x}_{t+1}, t+1)\right], \tag{16}$$

where $a_2$ is the coefficient for the second part of DPM-Solver++. The data prediction $\boldsymbol{x}_\theta(\boldsymbol{x}_t, t)$ and noise prediction $\boldsymbol{\epsilon}_\theta(\boldsymbol{x}_t, t)$ can be mutually derived from each other with equation 1.

$$\boldsymbol{x}_t = \alpha_t\boldsymbol{x}_\theta(\boldsymbol{x}_t, t) + \sigma_t\boldsymbol{\epsilon}_\theta(\boldsymbol{x}_t, t). \tag{17}$$

Therefore, we first convert the first order part $\boldsymbol{x}_\theta(\boldsymbol{x}_t, t)$ in equation 16 to noise prediction $\boldsymbol{\epsilon}_\theta(\boldsymbol{x}_t, t)$, and apply equation 8 to the converted $\boldsymbol{\epsilon}_\theta(\boldsymbol{x}_t, t)$, and finally convert it back to data prediction.

## 4 EXPERIMENTS

In this section, we show that our method can significantly boost the sampling quality and speed of existing solvers through extensive experiments. We employ FID and human preference model HPD v2 Wu et al. (2023) for comprehensive evaluation. All values are reported with 10k images unless specifically mentioned. We first present main results in Section 4.1 and provide more analyses in Section 4.3.

### 4.1 MAIN RESULTS

We illustrate the effectiveness of our method on few-step sampling with comprehensive experiments and analyses. Our experiments cover the main fast ODE samplers with different orders (1-order DDIM, 2-order DPM-Solver and DPM-Solver++), two model prediction modes (noise and date prediction), three general generation type (unconditional, class-conditional, and text-conditional), two existing guiding strategies (classifier-guided Dhariwal & Nichol (2021) and classifier-free guided Ho & Salimans (2022)), two main-stream sampling space (image space Dhariwal & Nichol (2021) and latent space Rombach et al. (2022)), main-stream datasets (LSUN-bedroom and ImageNet), as well as various guidance scales.

**Unconditional sampling.** We first compare the unconditional sampling quality of different methods on LSUN Bedroom Yu et al. (2015) and ImageNet Deng et al. (2009) datasets in Figure 3. The pre-trained diffusion models are from Dhariwal & Nichol (2021). Our method substantially boosts existing fast ODE solvers with better sampling quality and faster speed. The performance lift is especially obvious with fewer NFEs, which demonstrates the potential and effectiveness of DualFast for the practical deployment of generative diffusion models.

**Class-conditional sampling.** Besides the unconditional sampling, we adopt the class label as condition information. To this end, we employ the classifier-guided sampling strategy and the pre-trained models from Dhariwal & Nichol (2021). We validate the effectiveness of our method over baseline samplers under different guidance scale (s = 2.0 and 4.0). The results are shown in Figure 4. DualFast achieves consistent performance improvement with various solvers and guidance scales.

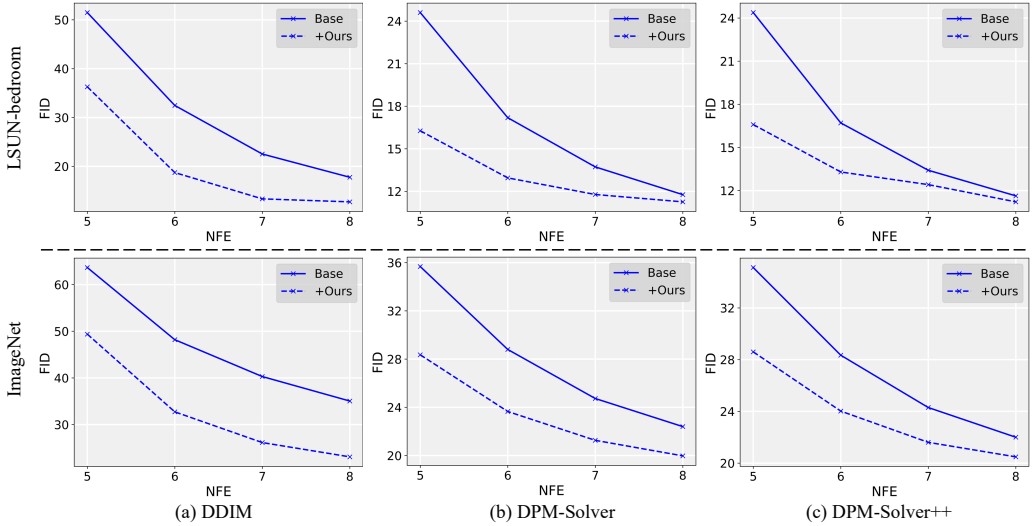

Figure 3: **Unconditional sampling results.** We compare DualFast with baseline samplers on LSUN Bedroom and ImageNet datasets. We report the FID ↓ of the methods with different NFEs. Experimental results show that DualFast is consistently better than baselines on pixel-space DPMs.

**Text-conditional sampling.** To further assess our method's performance across different condition types, we explore its application in a text-to-image stable diffusion model Rombach et al. (2022), which works in latent space utilizing classifier-free guidance strategy Ho & Salimans (2022). The guidance scale is set as 7.5 following the common setting. We sample the first 10K captions from the MS-COCO2014 validation dataset Lin et al. (2014) for input texts. Acknowledging the limitations of the FID metric in text-to-image scenarios Lu et al. (2022b); Zhao et al. (2023), and the inadequacy

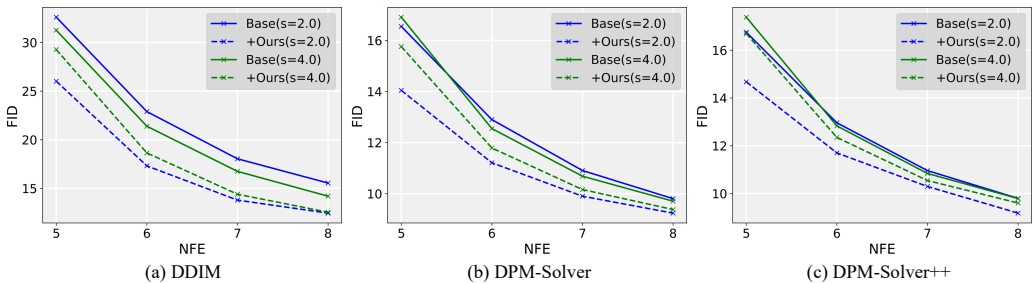

Figure 4: **Class-conditional sampling results.** Quantitative comparisons between existing samplers and our method with classifier-guided class condition, employing various classifier scales (s=2.0 and 4.0) and NFEs. DualFast achieves consistent performance improvement over baseline samplers.

of MSE for evaluating distribution convergence, we instead employ the HPD v2 Wu et al. (2023), a state-of-the-art model that predicts human preferences for images generated by text-to-image diffusion models. Results, illustrated in Figure 5, show our method consistently outperforms baseline samplers across varying orders, as evidenced by higher human preference scores.

## 4.2 VISUAL RESULTS

**Class-conditional sampling.** We provide a qualitative comparison between our method and previous sampling methods in Figure 6. We adopt NFE=7 with various classifier scales ($s = 0.0, 2.0,$ and $4.0$) for the presented samples. Our method consistently improves the image quality with better details, color, and contrast regardless of the samplers and guidance scales. For example, DualFast can even boost DDIM to achieve comparable visual results to DPM-Solver++.

**Text-conditional sampling.** Besides the pixel-space sampling results, we additionally provide visual results on stable diffusion in Figure 7. The NFE is set as only 5 to validate the performance bound of the compared methods. The classifier-free guidance scale is 7.5. Our method consistently generate more realistic images with fewer visual flaws and better structures than previous samplers. The above results illustrate that our method generalizes well to both pixel and latent space generation.

## 4.3 ANALYSES

In this section, we will provide more detailed analyses and ablation studies to further evaluate the effectiveness of DualFast. Due to page limit, **we leave more experiments and analyses in the supplementary material, including** the performance of DualFast on higher order (3-order UniPC), DiT Peebles & Xie (2023) architecture, higher guidance scale, larger NFEs, comparison with SDE-based samplers, performance upper bound of DualFast, sampling diversity metric, as well as more visual results.

**Ablation on the choices of $c$ and $\tau$.** DualFast has two main hyper-parameters within equation 8. For the mixing coefficient $c$, based on the prior that approximation error linearly decreases with step, we adopt a linearly decreasing strategy (from 0.5 to 0.0), which starts from 0.5 at step 0 and reaches 0.0 at step T. We also compare with constant mixing coefficient in Figure 8, where our linearly decreasing weight achieves higher performance than constant one. For the choice of step $\tau$, we investigate the performance of different $\tau$ in Figure 8. Higher $\tau$ usually corresponds to better performance. For simplicity and ease-of-use, we directly employ $\tau = T$. This means that we employ initial noise $\boldsymbol{x}_T$ as $\boldsymbol{\epsilon}_\theta(\boldsymbol{x}_\tau, \tau)$. Besides the above analyses, we additionally emphasise that due to our efficient paradigm, the choices of $c$ and $\tau$ are quite robust. Different choices all lead to substantial performance lift compared to the baseline solver.

**Reduced error.** We verify the effectiveness of DualFast with the common FID metric as well as numerical visual results. Besides, we also depict the MSE analysis in Table 1. Specifically, we adopt the results of 1000-step sampling as pseudo GT. Then we generate samples under various NFEs with both baseline solver and our DualFast. Since the NFEs and sampler orders are kept identical between the baseline solver and our DualFast, the discretization errors are also same bwtween these two pairs. Thus the additional MSE error reduction stems from less approximation error brought

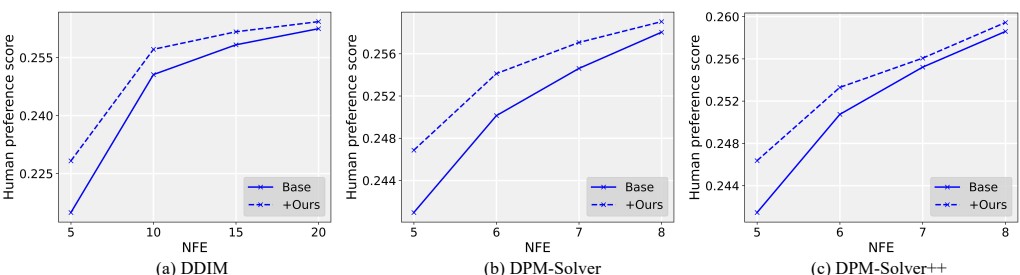

(a) DDIM      (b) DPM-Solver      (c) DPM-Solver++

Figure 5: **Text-conditional sampling results.** The results are reported on Stable diffusion model with the MS-COCO2014 validation dataset and guidance scale 7.5. We adopt the human preference score ↑ from HPD v2 Wu et al. (2023). DualFast performs better than previous baselines on latent-space DPMs.

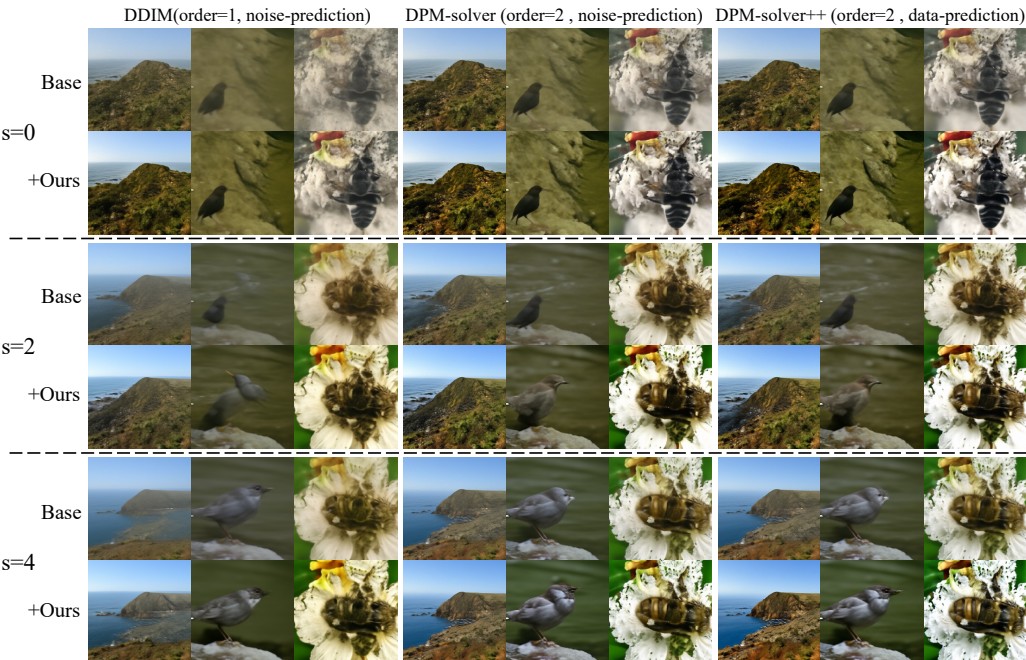

Figure 6: **Qualitative comparisons with class-conditional sampling in pixel-space.** All images are generated by sampling from a DPM trained on ImageNet 256×256 Dhariwal & Nichol (2021) with NFE= 7. The classifier scale $s$ is respectively set as 0.0, 2.0, and 4.0. Our method can generate more plausible samples with more visual details and higher contrast compared with previous samplers.

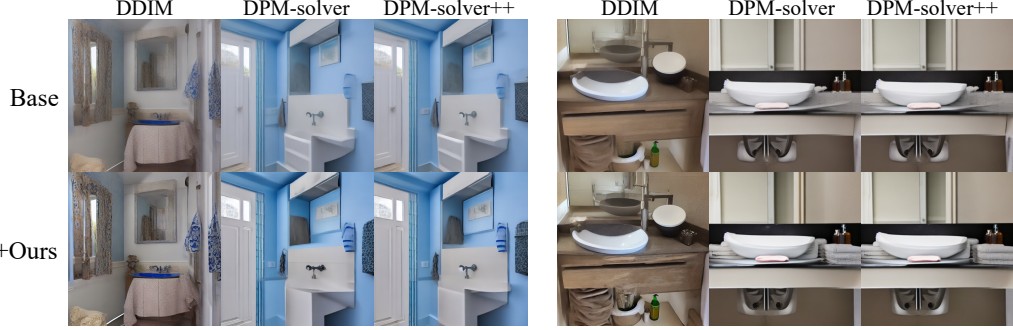

Figure 7: **Qualitative comparisons with text-conditional sampling in latent-space.** All images are generated with NFE = 5 and classifier-free guidance scale = 7.5. Our method can consistently generate more realistic images with fewer visual flaws than previous samplers of various orders.

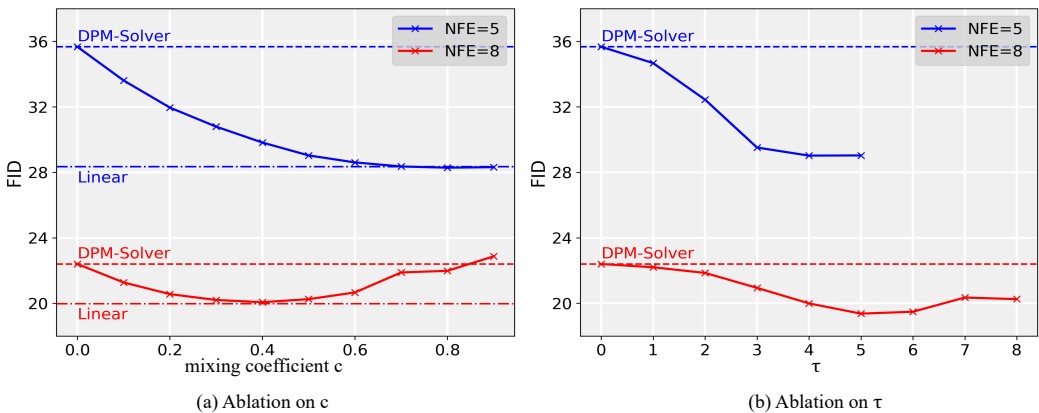

Figure 8: **Ablations on $c$ and $\tau$.** Due to our efficient paradigm, the choice of $c$ and $\tau$ are quite robust. Different choices all lead to substantial performance lift.(a) The adopted linear strategy achieves better performance than constant one. (b) Higher $\tau$ usually corresponds to better performance.

by our DualFast. More concretely, at each step $t$, the network prediction is modified with higher accuracy, thus contributing to smaller final error compared to the pseudo GT.

Table 1: **Reduced error.** MSE comparison between DPM-Solver and our method on various NFEs. Our method consistently reduces the MSE error and achieves further speedup than DPM-Solver.

| MSE($10^{-3}$) | NFE | | | | | | | | |
|---|---|---|---|---|---|---|---|---|---|
| | 5 | 6 | 7 | 8 | 9 | 10 | 12 | 15 | 20 |
| Base | 10.97 | 8.19 | 6.23 | 4.93 | 3.96 | 2.63 | 1.82 | 1.11 | 0.61 |
| Ours | 7.81 | 5.28 | 3.69 | 2.79 | 2.15 | 2.08 | 1.44 | 0.96 | 0.53 |

**Discussions and limitations.** Besides, despite the effectiveness of DualFast, it still lags behind training-based methods Salimans & Ho (2022); Song et al. (2023) with one-step generation. How to further close the gap between training-free methods and training-based methods requires future efforts.

## 5 CONCLUSION

We reveal that the sampling error in the generation process consists of two parts: discretization error and approximation error. Further, we propose a unified acceleration framework called DualFast for the fast sampling of DPMs by taking both errors into consideration to further accelerate sampling. We also verify the effectiveness of our method through extensive experiments.

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

## A    APPENDIX

This supplementary document is organized as follows:

Section B shows the concrete values in the quantitative comparison.

Section C shows experimental results on higher guidance scale.

Section D shows experimental results on larger NFEs.

Section E shows comparison with SDE-based sampler.

Section F shows the effectiveness of our method on DiT, a transformer-based diffusion model.

Section G explains the rationality of MSE for distribution measurement in diffusion models.

Section H explores the performance upper bound of the baselines and DualFast.

Section I shows the sampling diversity metric.

Section J shows the integration of our method into UniPC sampler.

Section K shows the detailed derivation of our method in DDIM sampler.

Section L depicts more visual results.

## B    QUANTITATIVE COMPARISON

In the main manuscript, we show the quantitative comparison with NFE-FID curve. In this part, we additionally present all comparisons in Table 2, 3 and 4. In Table 2, we depict the results of unconditional sampling with FID metric on LSUN Bedroom and ImageNet datasets. In Table 3, we show the results of class-conditional sampling with FID metric and different guidance scales. In Table 4, we show the results on text-conditional sampling with human preference score ↑, which is obtained from the sota human preference model HPD v2 Wu et al. (2023).

Table 2: Sample quality of unconditional sampling measured by FID ↓ on LSUN Bedroom and ImageNet datasets, varying the number of function evaluations (NFE).

| Dataset | Sampling Method \ NFE | 5 | 6 | 7 | 8 |
|---|---|---|---|---|---|
| LSUN Bedroom | DDIM(base) | 51.482 | 32.470 | 22.533 | 17.775 |
| | DDIM(+ours) | 36.288 | 18.738 | 13.352 | 12.744 |
| | DPM-Solver(base) | 24.607 | 17.191 | 13.722 | 11.766 |
| | DPM-Solver(+ours) | 16.270 | 12.947 | 11.776 | 11.257 |
| | DPM-Solver++(base) | 24.378 | 16.705 | 13.414 | 11.635 |
| | DPM-Solver++(+ours) | 16.594 | 13.295 | 12.416 | 11.218 |
| ImageNet | DDIM(base) | 63.653 | 48.191 | 40.295 | 35.047 |
| | DDIM(+ours) | 49.379 | 32.729 | 26.147 | 23.084 |
| | DPM-Solver(base) | 35.673 | 28.797 | 24.729 | 22.400 |
| | DPM-Solver(+ours) | 28.353 | 23.653 | 21.261 | 19.974 |
| | DPM-Solver++(base) | 35.118 | 28.342 | 24.298 | 22.003 |
| | DPM-Solver++(+ours) | 28.602 | 24.022 | 21.607 | 20.486 |

## C    HIGHER GUIDANCE SCALE

Guided sampling can significantly boost the sample quality compared to unconditional sampling. But high guidance scale would also cause the instability of the sampler Lu et al. (2022b) and poor sample quality. In this part, we show the results with guidance scale of 6.0 in Table 5, and avoid higher guidance scale. Compared with the results in Table 3, the sample quality is worse with guidance scale 6.0. But, DualFast can still consistently achieve better performance than the base sampler.

Table 3: Sample quality of class-conditional sampling measured by FID ↓ on ImageNet 256×256 Dhariwal & Nichol (2021), varying the number of function evaluations (NFE) and guidance scale.

| Guidance Scale | Sampling Method \ NFE | 5 | 6 | 7 | 8 |
|---|---|---|---|---|---|
| 2.0 | DDIM(base) | 32.599 | 22.894 | 18.043 | 15.552 |
| | DDIM(+ours) | 26.018 | 17.316 | 13.798 | 12.460 |
| | DPM-Solver(base) | 16.549 | 12.901 | 10.907 | 9.807 |
| | DPM-Solver(+ours) | 14.046 | 11.210 | 9.902 | 9.240 |
| | DPM-Solver++(base) | 16.753 | 12.962 | 10.960 | 9.799 |
| | DPM-Solver++(+ours) | 14.675 | 11.701 | 10.296 | 9.582 |
| 4.0 | DDIM(base) | 31.258 | 21.389 | 16.751 | 14.202 |
| | DDIM(+ours) | 29.265 | 18.645 | 14.401 | 12.554 |
| | DPM-Solver(base) | 16.909 | 12.548 | 10.685 | 9.707 |
| | DPM-Solver(+ours) | 15.759 | 11.790 | 10.163 | 9.380 |
| | DPM-Solver++(base) | 17.381 | 12.832 | 10.842 | 9.791 |
| | DPM-Solver++(+ours) | 16.695 | 12.350 | 10.546 | 9.605 |

Table 4: Sample quality of text-conditional sampling measured by human preference score ↑ (human preference model HPD v2 Wu et al. (2023)) with captions from MS-COCO2014 validation dataset, varying the number of function evaluations (NFE).

| Sampling Method \ NFE | 5 | 10 | 15 | 20 |
|---|---|---|---|---|
| DDIM(base) | 0.21492 | 0.25061 | 0.25831 | 0.26247 |
| DDIM(+ours) | 0.22828 | 0.25714 | 0.26168 | 0.26430 |
| Sampling Method \ NFE | 5 | 6 | 7 | 8 |
| DPM-Solver(base) | 0.24097 | 0.25014 | 0.25461 | 0.25803 |
| DPM-Solver(+ours) | 0.24687 | 0.25412 | 0.25707 | 0.25904 |
| Sampling Method \ NFE | 5 | 6 | 7 | 8 |
| DPM-Solver++(base) | 0.24146 | 0.25075 | 0.25521 | 0.25859 |
| DPM-Solver++(+ours) | 0.24637 | 0.25330 | 0.25607 | 0.25944 |

Table 5: Sample quality of class-conditional sampling measured by FID ↓ with guidance scale 6.0.

| Guidance Scale | Sampling Method \ NFE | 5 | 6 | 7 | 8 |
|---|---|---|---|---|---|
| 6.0 | DDIM(base) | 36.129 | 23.408 | 17.407 | 14.705 |
| | DDIM(ours) | 34.976 | 19.8449 | 14.3411 | 12.740 |

# D LARGER NFEs

In the main manuscript, we show that our method substantially elevates the sample quality in few-step sampling case (NFE<=8). In this part, we also validate the effectiveness of our method on larger NFEs in Table 6. DualFast improves the FID of DDIM from 28.906 to 20.559 when NFE=10, achieving two times acceleration (comparable to 20-step DDIM sampling).

Table 6: Sample quality of unconditional sampling on ImageNet dataset with larger NFEs.

| Sampler | DDIM(base) | | | DPM-Solver(base) | | | DDIM(ours) |
|---------|--------|--------|--------|--------|--------|--------|--------|
| NFE | 10 | 15 | 20 | 10 | 15 | 20 | 10 |
| FID | 28.906 | 22.781 | 20.344 | 21.250 | 20.004 | 19.533 | 20.559 |

# E COMPARISON WITH SDE-BASED SAMPLER

Large step size in stochastic differential equations (SDEs) violates the randomness of the Wiener process Kloeden & Platen (1992) and often causes non-convergence. Therefore, SDE-based sampler usually adopts hundreds of NFEs for inference. Certain methods Guo et al. (2023); Gonzalez et al. (2024) propose to accelerate SDE solvers but still require hundreds of steps for inference. Restart Xu et al. (2023) proposes to combine SDE and ODE via introducing stochasticity into the ODE process. These methods make promising attempts to accelerate SDE samplers, while still lag behind ODE solvers in speed. Besides, SDE-based sampler leads to stochastic generation, compared to the deterministic generation of ODEs. As shown in Fig. 9, we depict the visual comparison between these various samplers, including the SDE-based sampler DDPM Ho et al. (2020), the SDE-ODE combined sampler Restart Xu et al. (2023), as well as the ODE-based sampler DDIM Song et al. (2020a) and its enhanced version with our DualFast. DDPM suffers from blurry results with small NFEs. Restart effectively elevates the speed of DDPM but still generates low-quality images with small NFEs. Besides, both DDPM and Restart lead to stochastic generation. In contrast, DDIM sampler performs better with finer details and structure. Further, with our DualFast framework, the sampling quality and speed of DDIM are substantially boosted.

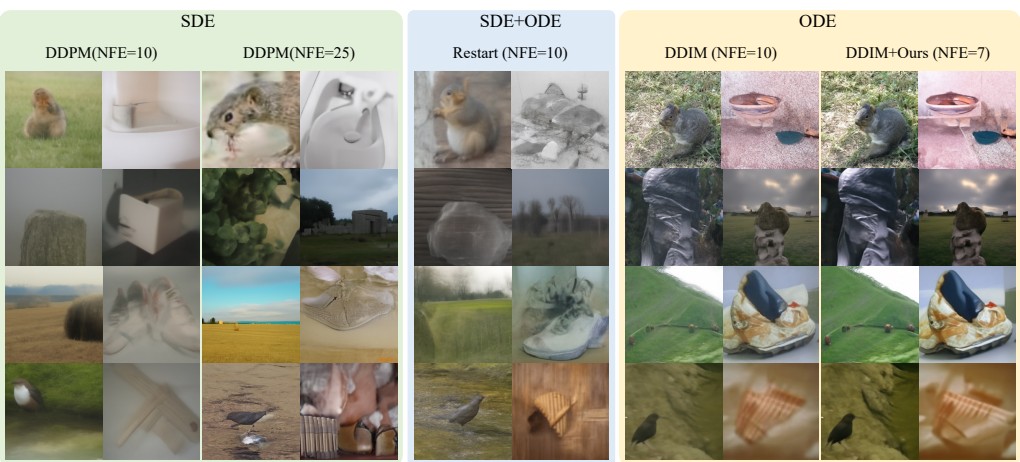

Figure 9: **Visual comparison with SDE-based samplers.** All images are generated by sampling from a DPM trained on ImageNet 256×256 Dhariwal & Nichol (2021). Our method is superior to both SDE and ODE samplers in quality and speed, generating more plausible samples with more visual details and higher contrast. Best viewed in color.

## F  PERFORMANCE ON DIT

In the main paper, we verify the effectiveness of our method on the two representative guided-diffusion Dhariwal & Nichol (2021) (class condition in pixel-space) and stable diffusion Rombach et al. (2022) (text condition in latent-space) models. In this section, we further demonstrate the efficacy of our method on DiT Peebles & Xie (2023), which adopts transformer Vaswani et al. (2017) architecture. Specifically, we adopt DiT-XL/2 with various guidance scales in Fig. 10. Our method significantly boosts the quality and speed of DDIM sampler, even achieving comparable visual results to DDIM of 50 NFEs with only 10 NFEs.

## G  MSE FOR DISTRIBUTIONS MEASUREMENT IN DIFFUSION MODELS

Adopting MSE to measure the distributions divergence in diffusion models is grounded with both theoretical guarantee and sufficient empirical support from classical and representative papers. (1) Employing MSE to measure the distribution divergence in this special case is theoretically guaranteed. Box & Tiao (2011) discusses MSE as a special case of maximum likelihood estimation when the error follows a Gaussian distribution. Murphy (2012) covers why MSE is a reasonable choice under the assumption of Gaussian noise. In the context of deep learning, LeCun et al. (2015) discusses the application of MSE, particularly in error measurement in generative models. Since in the context of diffusion models, gaussian distribution is the essential and default choice. MSE is thus a simple, reliable and rational metric to measure the distribution divergence, under the special case of gaussian distribution. (2) It is also a common practice of previous sampler pappers, that employing MSE to measure distribution distance. For example, the main-stream samplers (our baselines), including DPM-Solver++ Lu et al. (2022b) and UniPC Zhao et al. (2023), also employs MSE (l2 distance) to compare the convergence error between the results of different methods and 1000-step DDIM, in the text-to-image model provided by stable-diffusion. Besides, EDM Karras et al. (2022) focuses on the discretization error and also proposes to leverage root mean square error (RMSE) to measure the distribution distance between one Euler iteration and a sequence of multiple smaller Euler iterations, representing the ground truth.

## H  EXPLORING THE UPPER BOUND OF DUALFAST

It is important and of practical value to explore the performance upper bound of existing samplers and our DualFast for fast sampling. Concretely speaking, we desire to reveal the minimal sampling step required by existing samplers to generate visually clear and pleasing images. We adopt human preference as well as two well-known no-reference image quality assessment indicators: BRISQUE ↓ Mittal et al. (2012a) and NIQE ↓ Mittal et al. (2012b) to assess the visual results. As shown in Fig. 11, we depict the visual comparison under minimal sampling step of different samplers, and conclude two main conclusions. First, DualFast achieves a larger minimal step reduction than increasing sampler order. For example, 2-order DPM-Solver reduces the minimal step from 15 to 8, compared to 1-order DDIM. While, our DualFast enables DDIM to achieve minimal-step of 7.

Besides, DualFast can also significantly reduce the minimal step requirement of high order samplers, like DPM-Solver and DPM-Solver++. For example, DualFast further reduces the minimal sampling step of DPM-Solver from 8 to 6. This validates the generality and robustness of DualFast.

## I  SAMPLING DIVERSITY

We also investigate the diversity of the images generated by DualFast. In Table 7, we compare the sampling diversity of DualFast and base samplers with the inception score (IS) metric on ImageNet dataset. DualFast can consistently improve the sampling diversity on various NFEs.

## J  OURS-UNIPC

UniPC is the recent state-of-the-art high-order ODE solver. It employs a predictor-corrector framework, consisting of a predictor and a corrector. The unified corrector (UniC) can be applied after

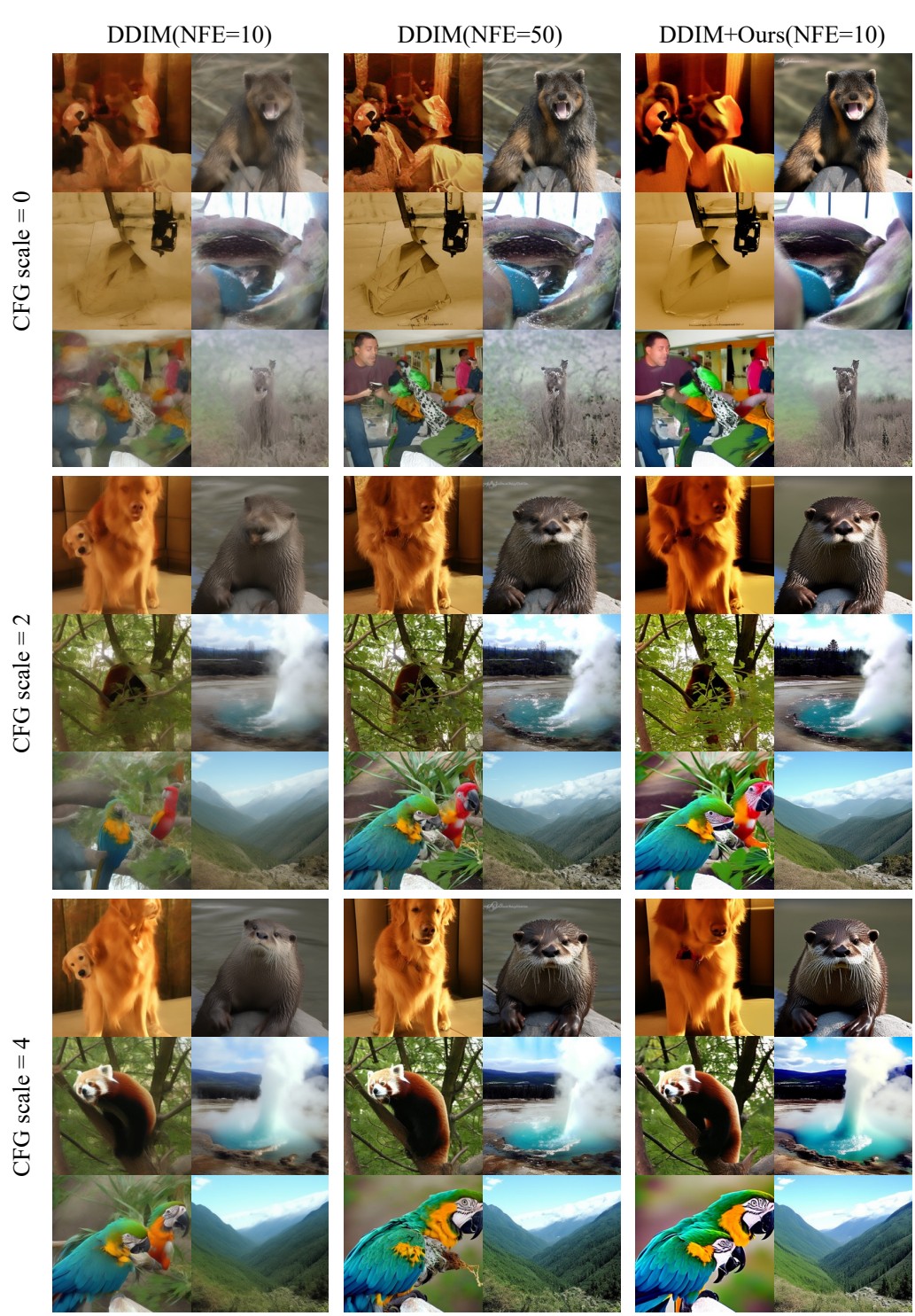

Figure 10: **Visual results on DiT with various guidance scales.** All images are generated by sampling from DiT-XL/2 Peebles & Xie (2023) with class condition. Our method significantly boosts the quality and speed of DDIM sampler, even achieving comparable visual results to DDIM of 50 NFEs with only 10 NFEs. Best viewed in color.

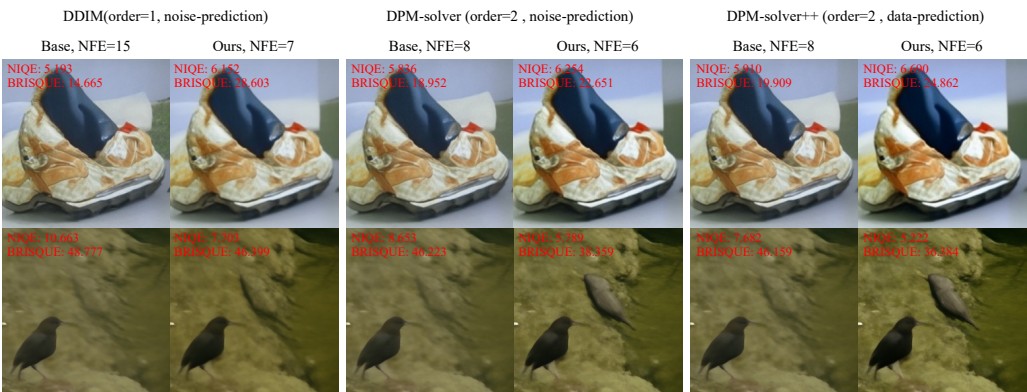

Figure 11: **The minimal steps required to generate visually clear images.** All images are generated unconditionally. Our method can consistently lower the minimal steps for clear image generation. Best viewed in color.

Table 7: **Comparisons of sampling diversity.** We compute the IS score on ImageNet dataset, where DualFast consistently improves the sampling diversity.

| IS | DDIM | | | | DPM-Solver | | | | DPM-Solver++ | | | |
|---|---|---|---|---|---|---|---|---|---|---|---|---|
| | 5 | 6 | 7 | 8 | 5 | 6 | 7 | 8 | 5 | 6 | 7 | 8 |
| Base | 29.9 | 40.3 | 46.8 | 52.6 | 54.5 | 61.8 | 67.5 | 72.5 | 54.5 | 62.1 | 68.7 | 71.3 |
| Ours | 38.7 | 54.4 | 64.2 | 69.9 | 61.6 | 70.0 | 74.3 | 75.4 | 58.6 | 67.1 | 71.4 | 72.7 |

any existing DPM sampler to increase the order of accuracy without extra model evaluations, and the unified predictor (UniP) supports arbitrary order. In this part, we take the most widely used third-order UniPC sampler as example. Concretely, the predictor in UniPC first gets an estimation of $\boldsymbol{x}_t^p$, then with the corresponding $\boldsymbol{D}_{t-1}$:

$$\boldsymbol{D}_{t-1}^{base} = \boldsymbol{\epsilon}_\theta(\boldsymbol{x}_t, t) + \sum_{m=0}^{p-2} a_m^p \left[ \boldsymbol{\epsilon}_\theta(\boldsymbol{x}_{t+m}, t+m) - \boldsymbol{\epsilon}_\theta(\boldsymbol{x}_{t+1+m}, t+1+m) \right], \tag{18}$$

where $p$ is the order of the predictor. Then the corrector in UniPC refines the estimation $\boldsymbol{x}_t^p$ with the corresponding $\boldsymbol{D}_t$:

$$\boldsymbol{D}_{t-1}^{\text{base}} = \boldsymbol{\epsilon}_\theta(\boldsymbol{x}_t, t) + \sum_{m=0}^{p-2} a_m^c \left[ \boldsymbol{\epsilon}_\theta(\boldsymbol{x}_{t+m}, t+m) - \boldsymbol{\epsilon}_\theta(\boldsymbol{x}_{t+1+m}, t+1+m) \right]$$
$$+ a_0^c \left[ \boldsymbol{\epsilon}_\theta(\boldsymbol{x}_{t-1}, t-1) - \boldsymbol{\epsilon}_\theta(\boldsymbol{x}_t, t) \right] \tag{19}$$

Then the modified version with our approximation error reduction strategy are presented as follows:

$$\boldsymbol{D}_{t-1}^{\text{base}} = \boldsymbol{\epsilon}_\theta(\boldsymbol{x}_t, t) + \sum_{m=0}^{p-2} a_m^c \left[ \boldsymbol{\epsilon}_\theta(\boldsymbol{x}_{t+m}, t+m) - \boldsymbol{\epsilon}_\theta(\boldsymbol{x}_{t+1+m}, t+1+m) \right]$$
$$+ a_0^c \left[ (1+c)\boldsymbol{\epsilon}_\theta(\boldsymbol{x}_{t-1}, t-1) - c\boldsymbol{\epsilon}_\theta(\boldsymbol{x}_\tau, \tau) - \boldsymbol{\epsilon}_\theta(\boldsymbol{x}_t, t) \right] \tag{20}$$

We also show the results on UniPC sampler with quantitative comparison in Table 8 and visual results in Figure 12. Our method can also boost the performance of this 3-order solver.

Table 8: Sample quality of unconditional sampling with UniPC sampler.

| Sampling Method \ NFE | 4 | 5 | 6 | 7 |
|---|---|---|---|---|
| UniPC(base) | 44.617 | 29.472 | 23.324 | 20.676 |
| UniPC(ours) | 42.066 | 28.367 | 22.882 | 20.451 |

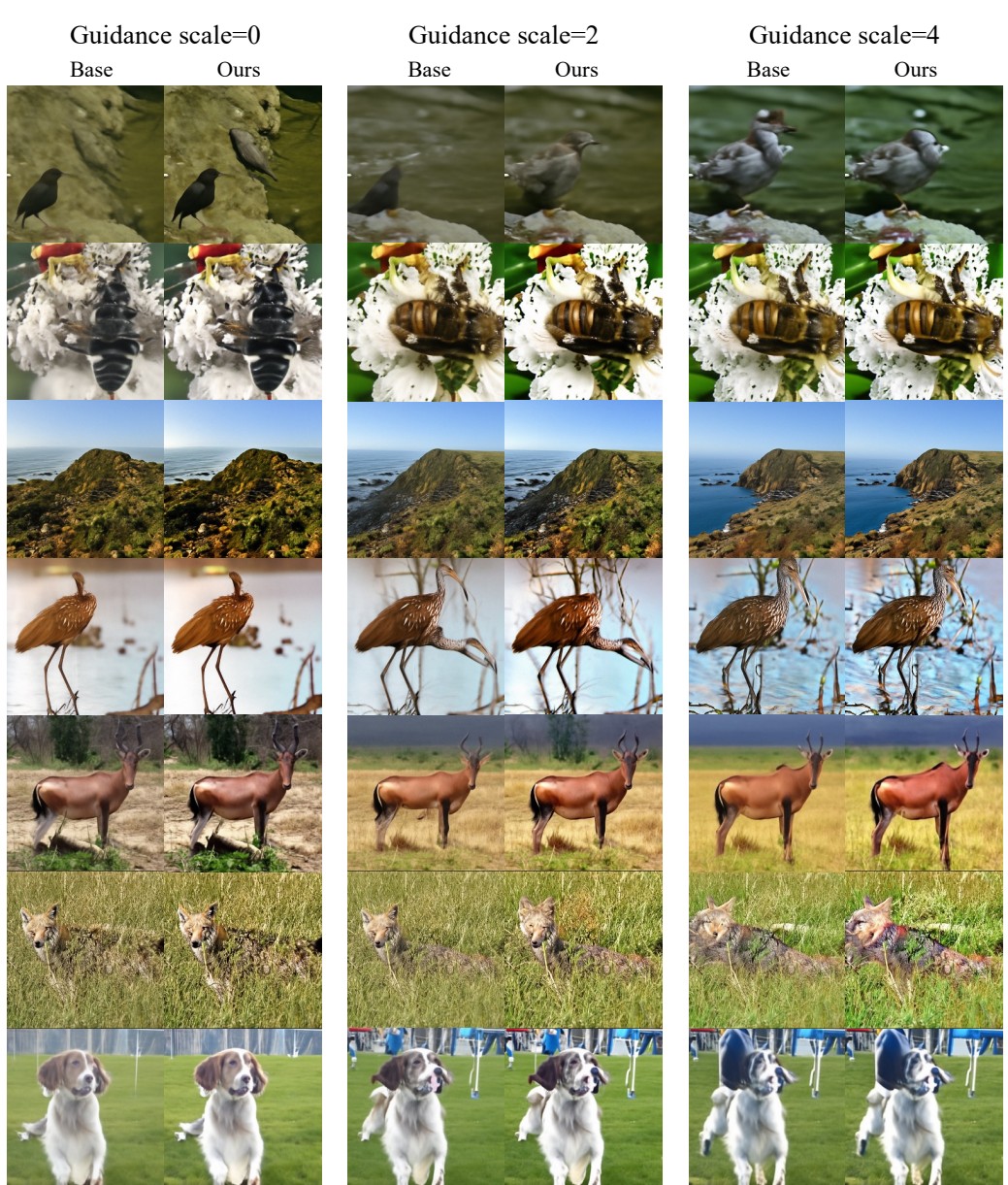

Figure 12: **Visual results of UniPC sampler on ImageNet dataset with various guidance scales.**

## K    DERIVATION IN OURS-DDIM

$$
\begin{aligned}
\boldsymbol{x}_t^{\text{ours}} &= a_t \boldsymbol{x}_0 + \sigma_t \boldsymbol{\epsilon}_\theta(\boldsymbol{x}_\tau, \tau) \\
&= \alpha_t \frac{\boldsymbol{x}_s - \sigma_s \boldsymbol{\epsilon}_\theta(\boldsymbol{x}_s, s)}{\alpha_s} + \sigma_t \boldsymbol{\epsilon}_\theta(\boldsymbol{x}_{\tau,\tau}) \\
&= \frac{\alpha_t}{\alpha_s} \boldsymbol{x}_s - \sigma_t \left( \frac{\alpha_t \sigma_s}{\sigma_t \alpha_s} - 1 \right) \boldsymbol{\epsilon}_\theta(\boldsymbol{x}_s, s) + \sigma_t \left[ \boldsymbol{\epsilon}_\theta(x_{\tau,\tau}) - \boldsymbol{\epsilon}_\theta(\boldsymbol{x}_s, s) \right] \\
&= \frac{\alpha_t}{\alpha_s} \boldsymbol{x}_s - \sigma_t (e^{h_t} - 1) \boldsymbol{\epsilon}_\theta(\boldsymbol{x}_s, s) + \sigma_t \left[ \boldsymbol{\epsilon}_\theta(\boldsymbol{x}_{\tau,\tau}) - \boldsymbol{\epsilon}_\theta(\boldsymbol{x}_s, s) \right] \\
&= \frac{\alpha_t}{\alpha_s} \boldsymbol{x}_s - \sigma_t (e^{h_t} - 1) \left\{ \boldsymbol{\epsilon}_\theta(\boldsymbol{x}_s, s) + \frac{1}{e^{h_t} - 1} \left[ \boldsymbol{\epsilon}_\theta(\boldsymbol{x}_s, s) - \boldsymbol{\epsilon}_\theta(\boldsymbol{x}_{\tau,\tau}) \right] \right\}
\end{aligned} \tag{21}
$$

Now, we can get the corresponding $\boldsymbol{D}_t^{ours}$ of the above equation.

$$
\begin{aligned}
\boldsymbol{D}_t^{ours} &= \boldsymbol{\epsilon}_\theta(\boldsymbol{x}_s, s) + \frac{1}{e^{h_t} - 1} \left[ \boldsymbol{\epsilon}_\theta(\boldsymbol{x}_s, s) - \boldsymbol{\epsilon}_\theta(\boldsymbol{x}_{\tau,\tau}) \right] \\
&= (1 + \frac{1}{e^{h_t} - 1}) \boldsymbol{\epsilon}_\theta(\boldsymbol{x}_s, s) - \frac{1}{e^{h_t} - 1} \boldsymbol{\epsilon}_\theta(\boldsymbol{x}_{\tau,\tau}),
\end{aligned} \tag{22}
$$

where $c = \frac{1}{e^{h_t} - 1}$ is the mixing coefficient. This is the general form. The coefficient in the main manuscript is slightly different from this form, and we will modify it to this general form in the revised version.

## L    MORE VISUAL RESULTS

In this part, we show more visual results with various samplers (DDIM, DPM-Solver, and DPM-Solver++), sampling types (unconditional, class-conditional, and text-conditional sampling), sampling spaces (pixel and latent space), NFEs (5, 6, 7, and 8), and guidance scales (0.0, 2.0, 4.0, and 7.5).

DDIM                              DPM-solver                          DPM-solver++

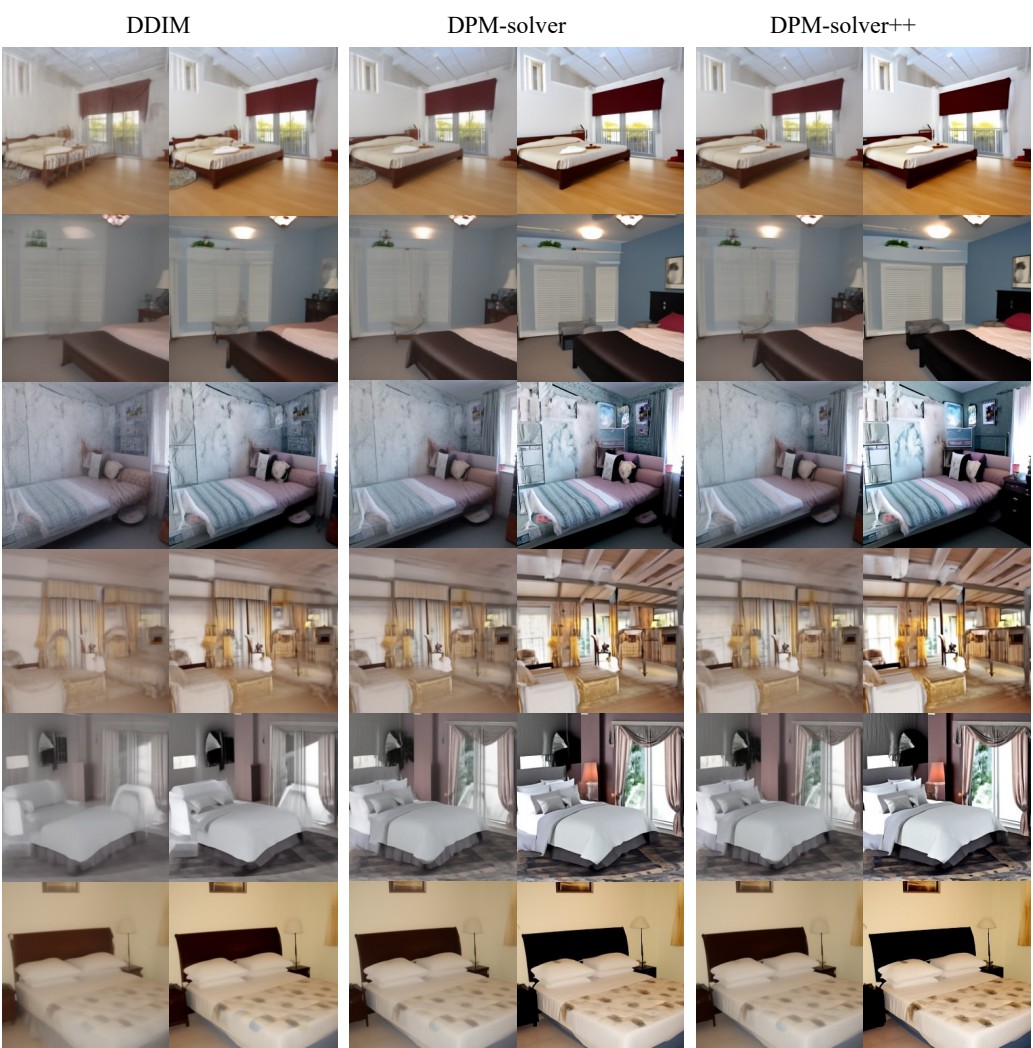

Figure 13: **Visual results of unconditional sampling on LSUN Bedroom dataset.**

DDIM                    DPM-solver                  DPM-solver++

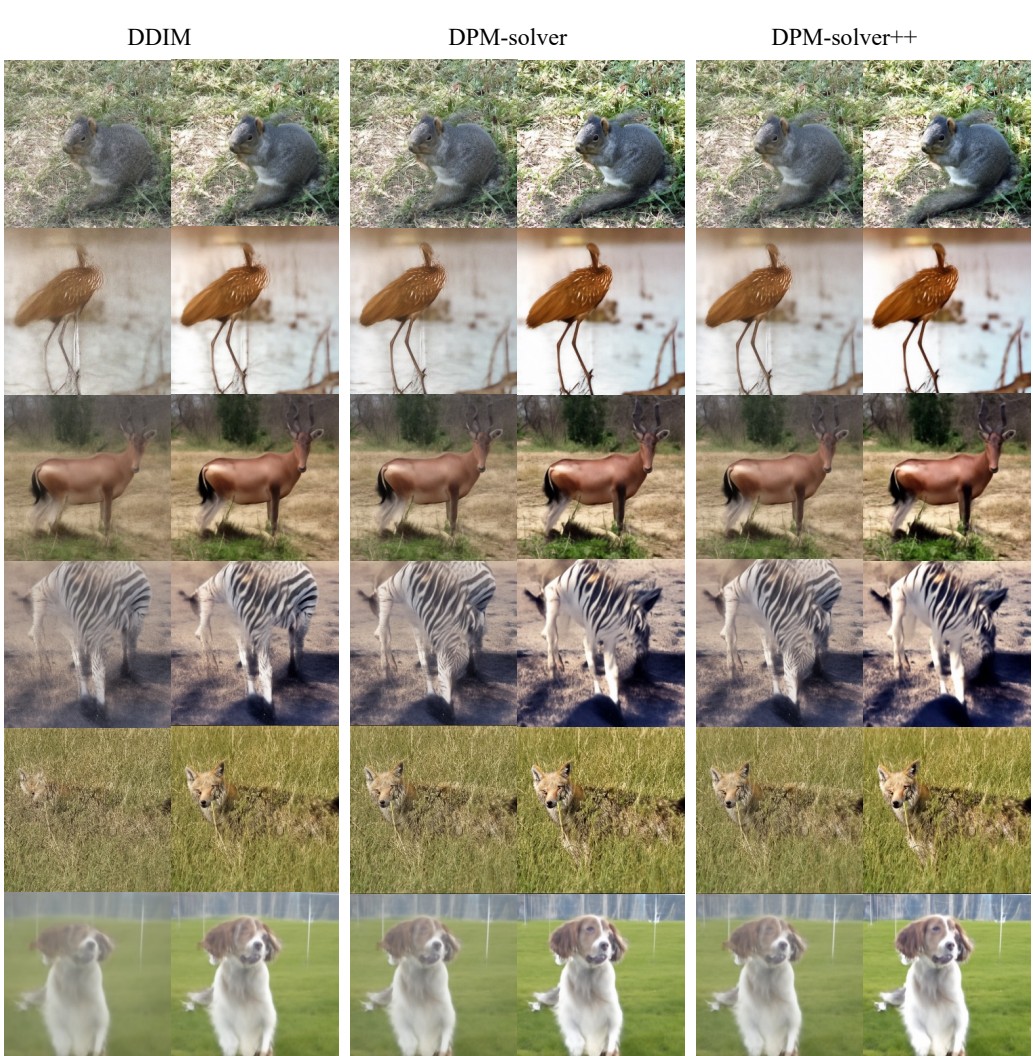

Figure 14: **Visual results of unconditional sampling on ImageNet dataset.**

1242
1243
1244
1245
1246
1247
1248
1249
1250
1251
1252
1253
1254
1255
1256
1257
1258
1259
1260
1261
1262
1263
1264
1265
1266
1267
1268
1269
1270
1271
1272
1273
1274
1275
1276
1277
1278
1279
1280
1281
1282
1283
1284
1285
1286
1287
1288
1289
1290
1291
1292
1293
1294
1295

| DDIM | DPM-solver | DPM-solver++ |

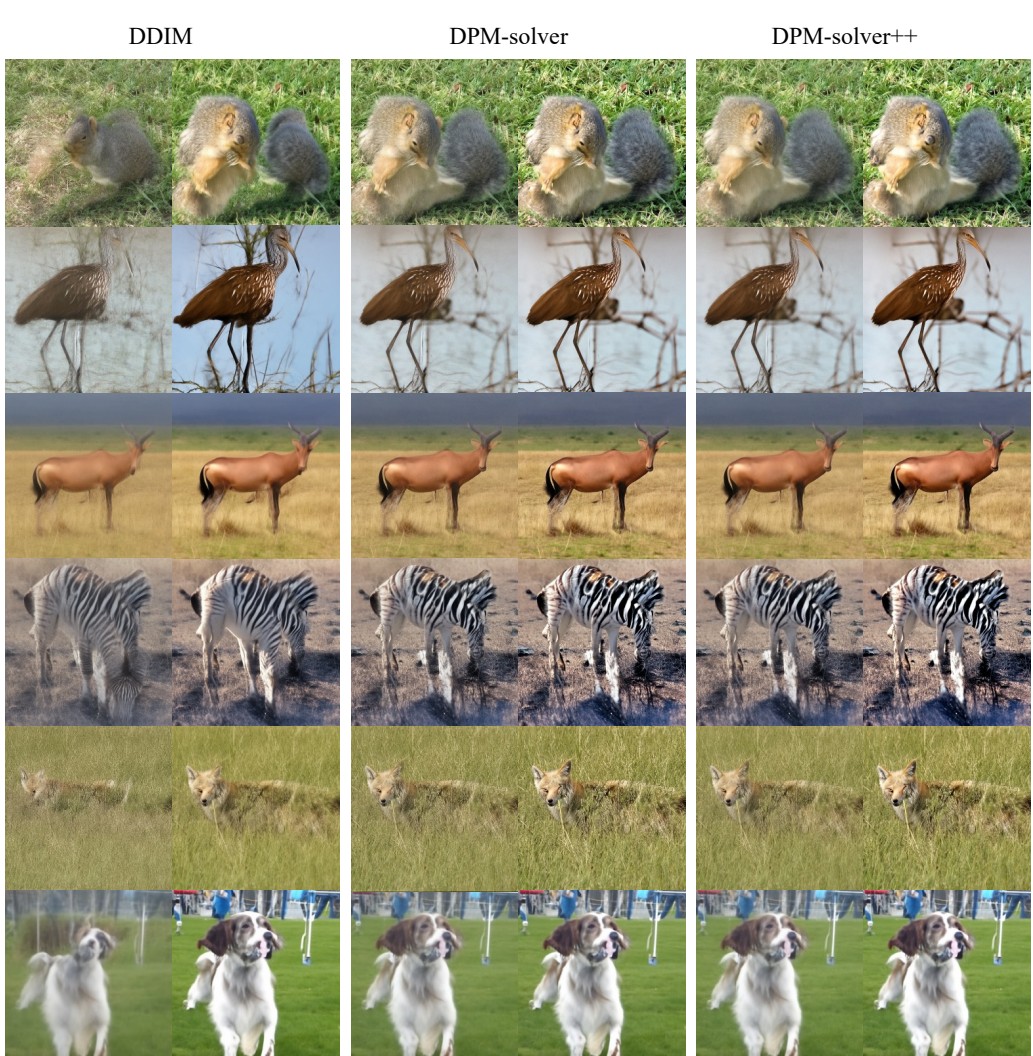

Figure 15: **Visual results of class-conditional sampling on ImageNet dataset with guidance scale 2.0.**

DDIM                    DPM-solver                    DPM-solver++

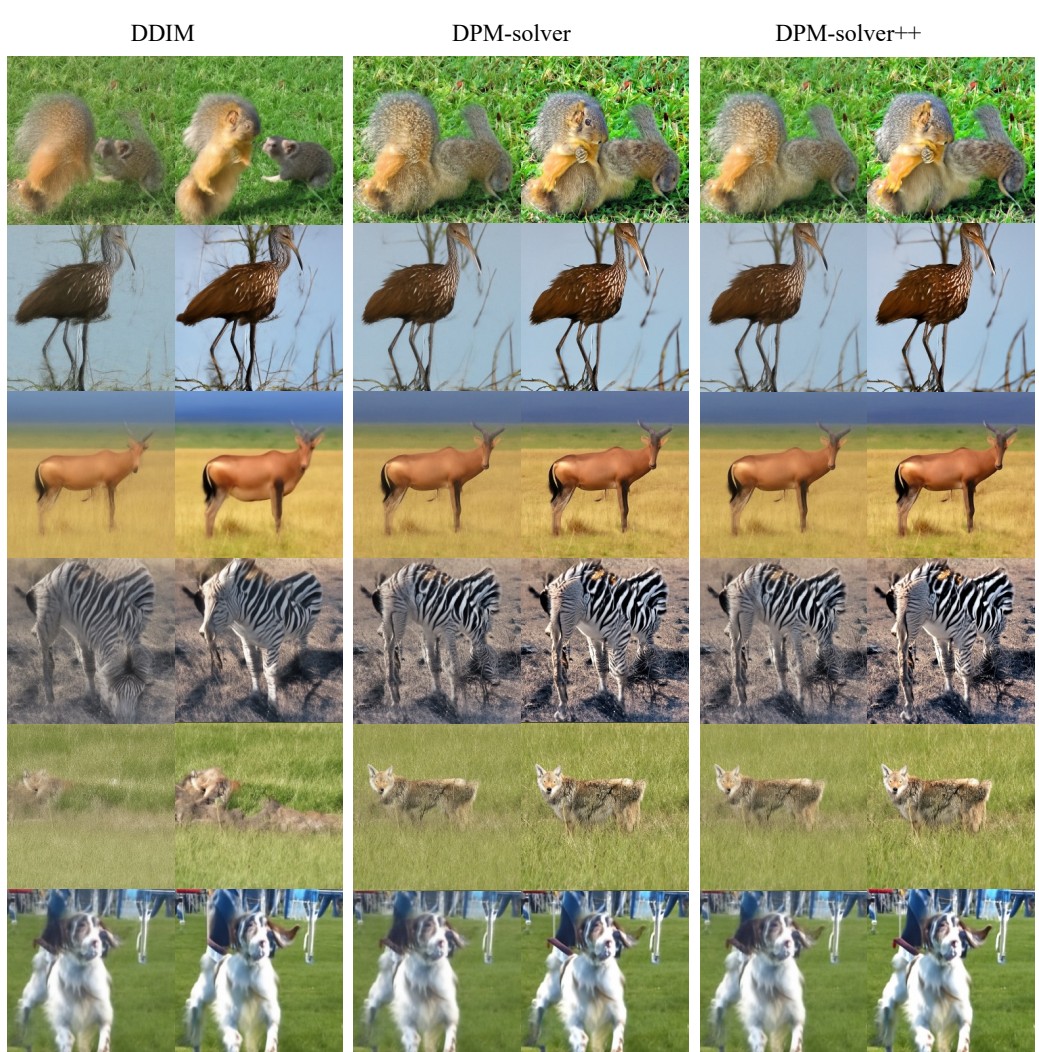

Figure 16: **Visual results of class-conditional sampling on ImageNet dataset with guidance scale 4.0.**

DDIM                    DPM-solver                    DPM-solver++

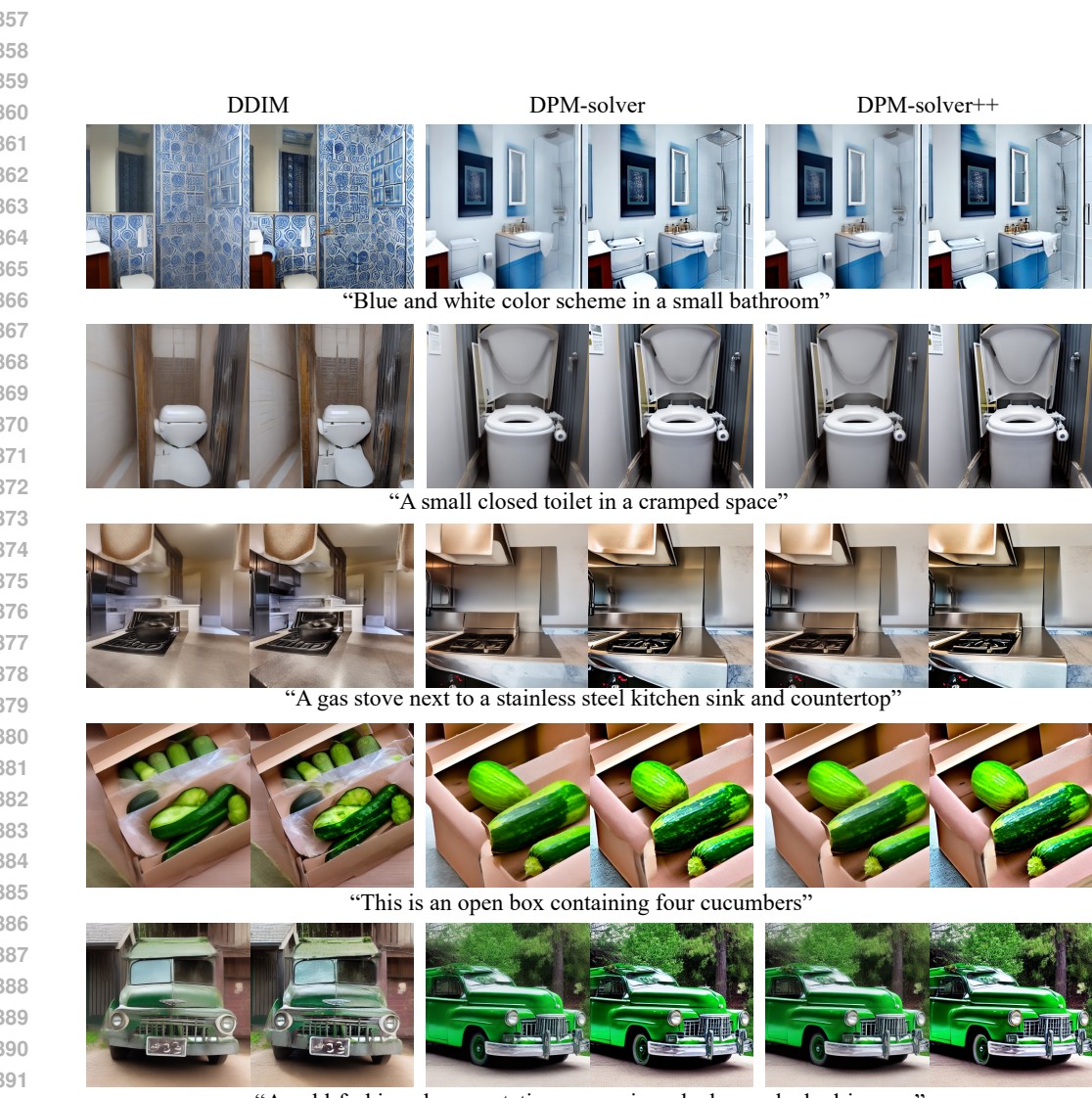

"Blue and white color scheme in a small bathroom"

"A small closed toilet in a cramped space"

"A gas stove next to a stainless steel kitchen sink and countertop"

"This is an open box containing four cucumbers"

"An old-fashioned green station wagon is parked on a shady driveway"

Figure 17: **Visual results of text-conditional sampling on stable diffusion with guidance scale 7.5.**

