# OpenReview forum: "DualFast: Dual-Speedup Framework for Fast Sampling of Diffusion Models"
_ICLR.cc/2025/Conference — Submitted to ICLR 2025_

### Official Review · Reviewer_u2p2 · 2024-10-21

**Soundness:** 1
**Presentation:** 3
**Contribution:** 2
**Rating:** 3
**Confidence:** 5

**Summary:**

This paper introduces a novel approach aimed at accelerating the sampling process in diffusion models. The proposed method seeks to minimize approximation errors by leveraging noise predictions from previous steps. When applied to DPM and DPM++ solvers, this approach effectively enhances the FID and human preference scores.

**Strengths:**

1. This paper primarily concentrates on reducing the approximation error, a factor overlooked by previous studies.
2. The experiments conducted in this paper are thorough.
3. This paper is well composed.

**Weaknesses:**

1. The paper do not illustrate the computation of the MSE. The analysis of the Dualfast model is based on the probability flow ODE, yet, as per Figure 2, the "exact form" is derived from the forward SDE. The "exact form" should be the analytical solution of the probability ODE, not the outcomes generated by the forward process of the diffusion model.

2. Line 263-269. The explanation of diffusion model is incorrect. According to [1], the expression for the optimal \epsilon_\theta can be computed explicitly. Given the training data $\{ y_1, …, y_N\}$, $\epsilon(x_t, t) = \sum_i \frac{\mathcal{N}(x_t; y_i, \sigma_t I)}{\sum_j \mathcal{N}(x_t; y_j, \sigma_t I)} \frac{x_t - \alpha_t y_i}{\sigma_t}$. This demonstrates that the optimal $\epsilon(x_t, t)$ is the weighted  average of the vectors pointing towards all the data points. The claim $\epsilon(x_\tau, \tau)$ is a better approximation of $\epsilon(x_t, t)$ can be contradicted with a simple 2D example with the situation where the training data only has two samples $\{(0,1), (0,-1)\}$.

3. Line 279. $x_\theta$ and $\epsilon_\theta$ are equivalent by equation (17), yet only the explicit $\epsilon_\theta(x_t, t)$ is replaced with $\epsilon(x_\tau, \tau)$. If $\epsilon(x_\tau, \tau)$ is indeed more accurate, the implicit part in $x_\theta$ should also be replaced.

4. Line 286. According to figure 8, the coefficient $c$ is non-negative.  However, the coefficient of $\epsilon_\theta(x_\tau, \tau)$ in equation (12) is non-positive, implying that $x_t$ is pushed further away from $\epsilon_\theta(x_\tau, \tau)$.  This contradicts the claim that $\epsilon_\theta(x_\tau, \tau)$ is closer to the ground truth.

5. The paper attributes the observed phenomena to a reduction in approximation error without sufficient evidence. It would be more beneficial to explore other potential causes, such as the possibility that the term $-c \epsilon_\theta(x_\tau, \tau)$  pushes $x_t$ away from 0, leading to increased brightness and saturation.


[1] Karras, Tero, et al. "Elucidating the design space of diffusion-based generative models." Advances in neural information processing systems 35 (2022): 26565-26577.

**Questions:**

1. Could you please elaborate on the methodology used to calculate the MSE?
2. As per [1], the optimal $\epsilon_\theta(x_t, t)$ can be computed for the 2D example with training data $\{(0,1), (0,-1)\}$. Could you compute and numerically compare $\epsilon_\theta(x_t, t)$ and $\epsilon(x_\tau, \tau)$?
3. Would you be able to substitute the implicit $\epsilon_\theta$ in $x_\theta$ and conduct experiments to demonstrate its improvements?

---

> ### Author Response · Authors · 2024-11-22
>
> **1: Computation of the MSE.**
> Thank for the valuable feedback. The mentioned analytical solution of the probability ODE exists theoretically but are prohibitive to calculate or employ in practical use. Actually, it is also impossible to exactly measure the distribution distance. The analyses in this paper provide a rough changing trend of the sampling errors, including adopting the MSE to approximately measure distribution distance. EDM [1] also employ similar way to measure the discretization error by calculating the MSE loss.
>
> [1] Karras, Tero, et al. "Elucidating the design space of diffusion-based generative models." Advances in neural information processing systems 35 (2022): 26565-26577.
>
> **2 : The explanation of diffusion model.**
> Actually, we know that the score function can be theoretically computed explicitly, and it may also be a common sense. While, for high-dimensional image data and large-scale training data, it is impossible to get the analytical solution, otherwise these is no sampling error. We understand that the reviewer hopes to be theoretically rigorous, while it is of limited practical meaning in most cases. Besides, we do not immediately understand how the mentioned toy example makes our explanation contradicted. Could you further cast on it. We are also grateful for these meticulous reviews.
>
> **3:  $x_\theta$ and $\epsilon_\theta$.**
> We only modify the $\epsilon_\theta$ part in equation with several reasons. First, equation 8 resembles the forward process $x_t = \alpha_t x_0 + \sigma_t \epsilon$, where $x_\theta$ estimates $x_0$ and $\epsilon_\theta$ estimates $\epsilon$. We thus only modify the $\epsilon_\theta$ part. Besides, this also elegantly results the $D_t$ of DDIM sampler from $\epsilon_\theta(x_t, t)$ to $(1+c) \epsilon_\theta(x_t, t) - \epsilon_\theta(x_\tau, \tau)$.
> From the performance perspective, we also find that this simpler implementation achieves better quality than modifying both $x_\theta$  and $\epsilon_\theta$ in equation 8.
>
> **4: Coefficient c.**
> Indeed, we do not agree that equation 12 pushes $x_t$ further away from $\epsilon_\theta(x_\tau,\tau)$ and causes contradiction. This can be simply verified through the comparison between equation 12 and equation 13. Equation 13 represents the $D_t$ expression of the 2-order sampler DPM-Solver. $\tau$ in equation 12 is also larger than t and reduces to equation 13 when $\tau=t+1$.
>
> **5: Other potential causes of approximation error.**
> Thanks for the suggestion. We are also open for other potential explanation. Combining response 4, another potential explanation may be that the implementation form of equation 12 is a more general form of previous high-order solvers and can well supplement them. Lastly, we want to emphasize that there may exist other intuitive explanation or theoretical proof. However, we reveal the significance of the approximation and integrate it into existing samplers with substantial performance improvement.

---

> > ### Comment · Reviewer_u2p2 · 2024-11-25
> >
> > Thank you for your clarification. However, there are still a few issues that need to be addressed.
> >
> > 1. Given the central role that the "exact form" plays in representing the approximation error, I was wondering if you could kindly provide a **step-by-step** method for calculating the $x_t$s of the exact form. Additionally, could you also provide the steps to calculate the MSE in Figure 2? This information could greatly assist readers in understanding how the approximation error is computed.
> >
> > 2. Let $\epsilon_ \theta$ and $\epsilon$ represent the output of the neural network and the ground truth, respectively. My primary concern lies in the assertion made in this paper, which states that $\epsilon_ \theta(x_T, T)$ is closer to $\epsilon(x_t, t)$ compared to $\epsilon_ \theta(x_t, t)$. Based on the analytical form of $\epsilon$, $\epsilon(x_T, T)$ significantly differs from $\epsilon(x_t, t)$. Given that calculating $\epsilon(x_t, t)$ is achievable for smaller datasets such as CIFAR-10, and considering the availability of open-source DDPM models on CIFAR-10, could you possibly provide quantitative results on CIFAR-10 to substantiate your claim?
> >
> > 3. If $\epsilon_\theta(x_\tau, \tau)$ is closer to $\epsilon(x_t, t)$ than $\epsilon_\theta(x_t, t)$, then using $\epsilon_\theta(x_\tau, \tau)$ to replace the $x_\theta$ term would result in a smaller error, and consequently, improved results. This seems to contradict your assertion in the rebuttal that "this simpler implementation achieves better quality than modifying both…". This suggests that the improvement in performance is not necessarily due to a reduction in the approximation error. Could you please clarify this?
> >
> > 4. In the DDIM scheme equation (8), we have $x_{t-1} = \frac{\alpha_{t-1}}{\alpha_t}x_t - (\sigma_t \frac{\alpha_{t-1}}{\alpha_t} - \sigma_{t-1})\epsilon(x_t, t)$. Given that $\alpha_t < \alpha_{t-1}$ and $\sigma_t > \sigma_{t-1}$, the coefficient of $\epsilon$ is negative. Indeed, since $\epsilon$ directs from $x_0$ to $x_t$, $-\epsilon$ accurately represents the denoting direction.
> > According to equation (10), this paper uses $x_{t-1} = \frac{\alpha_{t-1}}{\alpha_t}x_t - [(\sigma_t \frac{\alpha_{t-1}}{\alpha_t} - \sigma_{t-1})\epsilon(x_t, t) + \sigma_{t-1}(\epsilon(x_t, t) - \epsilon(x_\tau, \tau))]$. This, in turn, pushes the denoising direction away from $-\epsilon(x_\tau, \tau)$. Consequently, $\epsilon(x_\tau, \tau)$ assumes a role similar to the unconditional model in classifier-free guidance, providing a 'negative direction'.

---

> > > ### Author Response · Authors · 2024-11-25
> > >
> > > Thanks for the valuable feedback. Here, we depict more detailed explanations for addressing your concerns. While, frankly speaking, some of the explanations are already listed in the main manuscript.
> > >
> > > **Response to Q1.** In Line 200-241 and caption of Figure 2, we detailly introduce the calculation of Figure 2 and MSE. Here, we give step-by-step explanation for better understanding.
> > > Specifically, within the time interval [s, t], we construct three distinct transition processes, each subjected to varying levels of sampling error. 1) We sample plenty of images from the training data, forming the pristine image distribution $P(x_0)$. 2) For the first transition process (exact form) in Figure 2, we can derive the data distributions $P(x_s)$ and $P(x_t)$, employing the transition kernel outlined in equation 1. The transition from distributions $P(x_s)$ to $P(x_t)$ is free of both approximation and discretization errors.  3) For the second transition process (small step size) in Figure 2, starting from the same $P(x_s)$, we iteratively employ the denoising network to get $P(x_t^s)$. The transition from $P(x_s)$ to $P(x_t^s)$ can be seen as free from discretization error due to small step size, and only has approximation error. 4) For the third transition process (large step size) in Figure 2, starting from the same $P(x_s)$, we employ the denoising network once with large step size to get $P(x_t^l)$. The transition from $P(x_s)$ to $P(x_t^l)$ suffers from both discretization and approximation errors due to large step size. 5) Thus, we calculate the MSE loss between process 1 and 2, which represents the approximation error. The MSE loss between process 2 and 3 presents the discretization error.
> > >
> > > **Response to Q2.** We have introduced the rationality in Line 261-269. 1) As shown in Figure 2, the approximation error monotonically decreases as step t increases, indicating that the network estimation at smaller t suffers from more error. 2) For an intuitive explanation, at inference stage, the input of the denoising network at step T is pure Gaussian noise, and the network will also output the same noise pattern as the input. This means that this input Gaussian noise highly resembles the optimal output. However, as step t gets smaller, the noise level becomes lower, and the original noise pattern is harder to identify from the input. These explanations are also supported by the recent work [1]. This characteristic, where the network’s estimation is more desired at larger step, guides us to substitute the noise prediction at the current step t with that of a preceding, larger step τ .
> > >
> > > Hoping the above explanations address your concern. As for the claim to calculate the analytical form of $\epsilon$ on CIFAR-10 dataset, I’m really confused how to achieve this. In other words, if this is possible, the diffusion sampling on CIFAR-10 would be precisely accomplished, even not requiring samplers.
> > >
> > > [1] Yu H, Shen L, Huang J, et al. Unmasking Bias in Diffusion Model Training[J].
> > >
> > > **Response to Q3.** As we have replied in the last response, replacing the latter $\epsilon_\theta(x_t,t)$ part enjoys several benefits, including the similarity to the forward process, the elegant formulation, and the better experimental results. The reviewer may doubt part of the explanations, but should not neglect more other perspectives in the explanations. Besides, as for the doubt itself, replacing the $\epsilon_\theta(x_t,t)$ part is also within the principle of replacing the noise estimation, and should be viewed as contradiction.
> > >
> > > **Response to Q4.** For equation 8, the default and general way for expressing the sampling process is to reformulating it to equation 6. All the listed samplers in this paper can be formulated in equation 6, just with different $D_t$. We understand that the reviewer may want to understand equation 8 with other derivation, but it is not general and may not be helpful for understanding. For DDIM sampler in equation 8, its $D_t$ is $\epsilon_\theta(x_t,t)$. After our modification, the $D_t$ changes to $(1+c)\epsilon_\theta(x_t,t) – c\epsilon_\theta(x_\tau,\tau)$ as shown in equation 11. This formulation bears high similarity to other 2-order samplers, for example DPM-Solver in equation 13. This already verifies the rationality and effectiveness of our method,  as $\tau$ in equation 12 is also larger than t and reduces to equation 13 when $\tau$=t+1.

---

> > > > ### Comment · Reviewer_u2p2 · 2024-11-26
> > > >
> > > > Let's focus our discussion more precisely.
> > > >
> > > > Agreements:
> > > >
> > > > *A1*: The method proposed in this paper indeed enhances the quality of sampling.
> > > >
> > > > *A2*: A smaller approximation error will yield a better performance.
> > > >
> > > > Disagreements:
> > > >
> > > > *D1*: The value of $\epsilon_\theta(x_\tau, \tau)$ is closer to $\epsilon(x_t, t)$ than $\epsilon_\theta(x_t, t)$.
> > > >
> > > > *D2*: The improvement is attributed to the reduction of approximation error.
> > > >
> > > > **D1**
> > > >
> > > > The exact value of $\epsilon(x_t, t)$ can be efficiently calculated with CIFAR-10. The analytical expression can be found in equations (66)-(68) in EDM[1]. It is a weighted sum of all the training data, which is equivalent to the expression of $\epsilon(x_t, t)$ in weaknesses 2 in the previous review section. With the analytical form and a trained neural network, we are able to calculate the approximation error in a more direct and accurate manner. Could you possibly carry out this experiment and provide the numerical results?
> > > >
> > > > We train a neural network instead of using the analytical expression because when the number of training data is large, calculating the weighted sum over all data becomes inefficient. A neural network is utilized to obtain the weighted sum in only one NEF.
> > > >
> > > >
> > > > **D2**
> > > >
> > > > Would you agree with the statement below? Please respond directly.
> > > >
> > > > *S1*: When calculating the right-hand side of equation (10), since we do not have $x_\theta$, we need to convert it to $\epsilon_\theta$. As $\epsilon_\theta(x_\tau, \tau)$ is closer to $\epsilon(x_t, t)$ than $\epsilon_\theta(x_t, t)$ according to *D1*, replacing $x_\theta$ with $\epsilon_\theta(x_\tau, \tau)$ will result in an even smaller approximation error on the right-hand side of equation (10).
> > > >
> > > > If you agree with *S1*, replacing $x_\theta$ results in a smaller approximation error and will yield better results according to *A2*. However, this creates a contradiction. Could you please resolve this contradiction directly with out involve “other perspectives”?
> > > >
> > > > Replacing $\epsilon(x_t, t)$ improves the performance, which is a necessary condition of *D2* but not sufficient. Hence, replacing $\epsilon(x_t, t)$ with $\epsilon_\theta(x_\tau, \tau)$ to enhance performance does not contradict the statement "The improvement is NOT due to the reduction of approximation error."
> > > >
> > > >
> > > > References
> > > >
> > > > [1] Karras T, Aittala M, Aila T, et al. Elucidating the Design Space of Diffusion-Based Generative Models. NeurIPS 2022.

---

> > > > > ### Author Response · Authors · 2024-11-28
> > > > >
> > > > > Looking through the whole discussion, from the initial several weakness points to the current two specific disagreements, does the reviewer acknowledge that part of his concerns are addressed? This is important for fairly evaluating the contribution of our paper, since the reviewer seems to hold a very negative view to this work. Below we list the detailed explanations for your remaining concerns.
> > > > >
> > > > > **Reply to D1.**
> > > > > We carefully examine the equations mentioned by the reviewer. 1) The equation (66)-(68) as well as the expression of the denoiser in equation (57) are all derived from the prerequisite of variance exploding (VE), while our paper in conducted under the variance preserving (VP) setting. Besides, we also get confused on the equation of $\epsilon(x_t, t)$ in weakness 2 of previous review section. It seems that that equation blends VP and VE improperly. Even ignoring this error, this equation should be $ \epsilon\left(x_{t}, t\right)=\frac{x_t}{\sigma_t} - \sum_{i} \frac{\mathcal{N}\left(x_{t} ; y_{i}, \sigma_{t} I\right)y_i}{\sum_{j} \mathcal{N}\left(x_{t} ; y_{j}, \sigma_{t} I\right)} \frac{\alpha_t}{\sigma_t}$. 2) This equation is also consistent with our method. For example, when t is large and approaches T, $D(x, \sigma)$ represents the dataset mean , $\sigma_t \sim 1$, and $ \frac{\alpha_t}{\sigma_t} \sim 0$, which reduces $ \epsilon\left(x_{t}, t\right)$ to be $x_T$. This also verifies the rationality of replacing the noise estimation with $x_T$.
> > > > >
> > > > > **Reply to D2.**
> > > > > We treat the operation in your statement as an optional way, but not necessity. For DDIM sampler, our adopted way in the paper identically equals to replacing the  noise estimation $\epsilon_\theta(x_t,t)$ with $(1+c)\epsilon_\theta(x_t,t) – c\epsilon_\theta(x_\tau,\tau)$.  This equation resembles the 2-order DPM-Solver sampler in equation 13, also indicating the effectiveness of out method. The intuition for modifying $\epsilon_\theta$ is simple and straightforward. Equation 8 is a combination of data ($x_\theta$) and noise ($\epsilon_\theta$), it is natural to only replace the latter $\epsilon_\theta$ part. $x_\theta$ is viewed independently as the approximation of $x_0$. Some may hold that $x_\theta$ is implicitly correlated with $\epsilon_\theta$, via $x_\theta(x_t,t) = x_t/\alpha_t - \sigma_t/\alpha_t \epsilon_\theta(x_t,t)$. While, due to the existence of coefficient $ \sigma_t/\alpha_t $, slight modification of $ \epsilon_\theta(x_t,t)$ is amplified, significantly affecting the approximation of $x_0$.

---

> > > > > > ### Comment · Reviewer_u2p2 · 2024-11-29
> > > > > >
> > > > > > Since the primary point of contention is *D1*, I believe we should strive to reach some form of consensus before we proceed with the discussion.
> > > > > >
> > > > > > Now, let's address the statement
> > > > > >
> > > > > > *S1*: The analytical value of $\epsilon$ of VP SDE can be calculated.
> > > > > >
> > > > > > Here are the derivation steps:
> > > > > >
> > > > > > 1. Data Distribution: $p_{data} = \frac{1}{Y} \sum_{i=1}^Y \delta(x - y_i)$
> > > > > > 2. Marginal Distribution for $x_t$: Given $x_t = \alpha_t x_0 + \sigma_t \epsilon$, we find $p(x_t, t) = \int_{\mathbb{R}^d} p_{data}(x_0) p(x_t | x_0) d x_0 = \frac{1}{Y} \sum_{i=1}^Y \mathcal{N}(x_t; \alpha_t y_i, \sigma_t^2 I)$
> > > > > > 3. The Score Function and $\epsilon$: $\epsilon(x_t, t) = -\sigma_t \nabla_{x_t} \log p(x_t, t) = \sum_i \frac{\mathcal{N}(x_t; \alpha_t y_i, \sigma_t I)}{\sum_j \mathcal{N}(x_t; \alpha_t y_j, \sigma_t I)} \frac{x_t - \alpha_t y_i}{\sigma_t}$
> > > > > > 4. Equivalent Form: Noting that $\sum_i \frac{\mathcal{N}(x_t; \alpha_t y_i, \sigma_t I)}{\sum_j \mathcal{N}(x_t; \alpha_t y_j, \sigma_t I)} = 1$, $\epsilon(x_t, t)$ can also be expressed as $x_t - \sum_i \frac{\mathcal{N}(x_t; \alpha_t y_i, \sigma_t I)}{\sum_j \mathcal{N}(x_t; \alpha_t y_j, \sigma_t I)} \frac{\alpha_t y_i}{\sigma_t}$.
> > > > > >
> > > > > > Although I missed the $\alpha_t$ in previous expressions, this omission does not affect the establishment of *S1*.
> > > > > >
> > > > > > Given that these four steps are solely mathematical derivations, they are either correct or incorrect. Could you please confirm whether you agree with the above derivation? Please respond with a simple **Yes** or **No**.
> > > > > >
> > > > > > If you agree, could you provide numerical results on CIFAR-10 (or other datasets) for comparison between $\epsilon(x_t, t)$, $\epsilon_\theta(x_t, t)$, and $\epsilon_\theta(x_\tau, \tau)$?
> > > > > >
> > > > > > However, if you disagree, could you please identify the step in the previous derivation that you believe to be incorrect?

---

> > > > > > > ### Author Response · Authors · 2024-12-02
> > > > > > >
> > > > > > > Thank you very much for continued attention to our work and for the valuable feedback provided! We are delighted that most of your concerns are addressed. For the last remaining concern in D1, we mask substantial effofts in both the theoretical derivation and experiments as follows.
> > > > > > >
> > > > > > > **The derivation.**
> > > > > > > We agree with the first two steps in your derivation and they are also consistent with the derivations in EDM [1]. For step 3 and 4, we do not follow up the transition and also find it deviate from the derivation in EDM. Specifically, the ideal denoiser should be $ D\left(x_{t}, t\right)= \sum_{i} \frac{\mathcal{N}\left(x_{t} ; \alpha_t y_{i}, \sigma_{t} I\right)y_i}{\sum_{j} \mathcal{N}\left(x_{t} ; \alpha_t y_{j}, \sigma_{t} I\right)} $, aligning with equation 57 in EDM. Then following the DDIM sampler of $x_t=\alpha_t x_0+ \sigma_t \epsilon$, the noise estimation should be $\epsilon(x_t,t)=\frac{x_t}{\sigma_t} - \sum_{i} \frac{\mathcal{N}\left(x_{t} ; \alpha_t y_{i}, \sigma_{t} I\right)y_i}{\sum_{j} \mathcal{N}\left(x_{t} ; \alpha_t y_{j}, \sigma_{t} I\right)} \frac{\alpha_t}{\sigma_t}$. This differs from yours derivation in the coefficient of $x_t$.
> > > > > > >
> > > > > > > **The experiments.**
> > > > > > > We review the available codes and checkpoints on CIFAR10 dataset, and identify iDDPM[2] as our ideal codebase, which works in VP and noise prediction. We first calculate the ideal denoiser $ D\left(x_{t}, t\right)$ and derive the corresponding $\epsilon(x_t,t)$. Then we measure the MSE distance between 1) $\epsilon(x_t,t)$ and $x_T$; and 2) $\epsilon(x_t,t)$ and $\epsilon_\theta(x_t,t)$. By default, we adopt 20 steps for DDIM sampler. The calcalution is conducted on several samples with average to avoid randomness.
> > > > > > >
> > > > > > > | MSE      | t=19 | t=18 | t=17 | t=16  | t=15 | t=14 | t=13 | t=12  | t=11 | t=10 |
> > > > > > > | :---        | :----: | :----: | :----: | :----: | :----: | :----: | :----: | :----: | :----: | :----: |
> > > > > > > | $\epsilon(x_t,t)$ and $x_T$      |  0.0020  |  0.0082   | 0.021  | 0.039   | 0.065  |  0.104  |  0.156 |  0.227  |  0.324 |  0.456  |
> > > > > > > | $\epsilon(x_t,t)$ and $\epsilon_\theta(x_t,t)$    | 0.0025   |  0.0101  | 0.026  |  0.049  | 0.083  |  0.132  | 0.196  |  0.279  |  0.393 |  0.541  |
> > > > > > >
> > > > > > > The experimental results align with our claim that the approximation error increases as step t decreases and that the value of
> > > > > > > $\epsilon(x_t,t)$ is closer to $x_T$ than $\epsilon_\theta(x_t,t)$. Note that the results on $t<10$ are also consistent with our conclusion, we save it due to pape size. Besides, as we all realized, the calculation in such way is much slower than the ddim sampler.
> > > > > > >
> > > > > > > [1] Karras T, Aittala M, Aila T, et al. Elucidating the Design Space of Diffusion-Based Generative Models. NeurIPS 2022.
> > > > > > >
> > > > > > > [2] Nichol A Q, Dhariwal P. Improved denoising diffusion probabilistic models[C]//International conference on machine learning. PMLR, 2021: 8162-8171.

---

### Official Review · Reviewer_heMq · 2024-10-28

**Soundness:** 3
**Presentation:** 2
**Contribution:** 4
**Rating:** 6
**Confidence:** 4

**Summary:**

The authors consider the problem of accelerating DPM sampling using training-free methods. They highlight the fact that prior works in this area focus only on discretization error, neglecting the demonstrated substantial impact of approximation error in the total error. The main motivation, demonstrated in Figure 2, is that the approximation error of DPM networks is worst at small noise levels (i.e., close to x_0 along the probability path). This motivates the use of higher-noise-level outputs in the update steps of ODE solvers for DPM generation, where the goal is to reduce approximation error in addition to discretization error (where the latter is provided by existing efficient DPM solvers).

**Strengths:**

The method addresses a highly important issue (approximation error) in DPM sampling which they note is neglecting in prior works. Moreover, they demonstrate that this issue may have a greater impact than discretization error. Therefore, this is a highly relevant problem.

The training-free approach to mitigate approximation error is a good contribution, especially given the fact that this approach can be used in a plug-and-play manner with many DPM sampling techniques addressing discretization error.

Finally, the authors provide comprehensive experiments in a wide variety of settings and sampling schemes. This demonstrates the robustness of their method.

**Weaknesses:**

Some aspects of the background and method are not very clear. For instance, in equations 8 and 10, are both x_\theta and \epsilon_\theta outputs of the neural network? It is not immediately clear to me how this improves the approximation error of the neural network. I think the main contribution should be made more clear, or at least, it should be more clear in the main text how the modified sampling scheme leads to decreased approximation error.

I think the order of Sections 2.1 and 2.2 is a bit confusing. Equation 6 seems to be the “base” equation from which all solvers can be specified - maybe it should be defined earlier. To be more clear, maybe the “ODE-based sampling process” can be included in Section 2.2, which focuses on sampling.

Formatting Issues:

*The citation style makes it difficult to read the paper, especially the introduction. In-text citations should be surrounded by parentheses.

*Line 12: “While, they suffer from…” -> “However, they suffer from…” (while does not make sense in this context)

*Line 14: “introduces” -> “introduce”

*Line 24: “boost” -> “boosts”

*Lines 116-119: it could be necessary to comment on how \sigma_t and \alpha_t are defined

*Line 166: “while ignore” -> “while ignoring”

*Line 271: “general from applicable” I’m not sure what this means

*Line 280: “of equation 11” -> “of equation 9”

*Line 284: “between equation 9 and equation 9” -> “between equation 9 and equation 11”

*Line 289-291: The two uses of “Besides” in this paragraph are confusing. I think they can be removed in both sentences.

*Line 299/300: Missing period after “Ours-DPM-Solver”

*Line 311/312: Missing period after “Ours-DPM-Solver++”

*Line 311/312: “DPM-Solver++ is the sota and default samplers in stable diffusion model” -> “DPM-Solver++ is the SOTA and the default sampler for the stable diffusion model.” Also need a citation here.

*Line 430/431: “bwtween” -> “between”

*Line 514/515: “Besides” doesn’t make sense here

**Questions:**

Figure 2: It is mentioned that the time interval is divided into 9 time periods of size 111 (suggesting NFE=9), but also that NFE=111 and NFE=1. Also, if NFE=1 for operation 3, how do you obtain multiple data points for MSE in the right-side figure?

Figure 2 (right): Is T=1000 the initial time (x_t’ = x_s) and time T=0 the final time (x_t’ = x_t)? Maybe the x-axis should be inverted to demonstrate that error increases as the sampling procedure is run for a larger number of steps. Maybe the x-axis label should be something other than t, since it is suggested to be a fixed time point in the left side figure.

It is not clear to me how noise prediction/data predictions models can be derived from each other (the end of section 3). Is it possible to convert a noise prediction DPM to a data prediction DPM?

---

> ### Author Response · Authors · 2024-11-22
>
> **1: Main contribution more clear.**
> Equation 8 is the sampling formulation of DDIM sampler, where $x_\theta$  is the estimation of $x_0$ and $\epsilon_\theta$ is the estimation of $\epsilon$. In this paper, the denoising network predicts the noise, and the noise prediction and data prediction can be mutually converted as detailed in response 5. This sampling formulation resembles the forward process $x_t = \alpha_t x_0 + \sigma_t \epsilon$.
> How to reduce approximation error? In Figure 2 and L260-269, we show that the approximation error decreases as the step t increases, and the network’s estimation is more desired at larger step. This motivates us to substitute the noise prediction at the current step t with that of a preceding larger step $\tau$. Thus, in eq 10, we replace the noise estimation part $\epsilon_\theta(x_t, t)$ with $\epsilon_\theta(x_\tau, \tau)$.
>
> **2: Order of Sections 2.1 and 2.2**
> Much thanks for the suggestion. It is reasonable to combine “ODE-based sampling process”  part and Section 2.2 for easier understanding. We will modify this in the revised version.
>
> **3: Minor typos.**
> Thank for the careful review. We will modify the citation format and correct these typos in the revised version.
>
> **4: NFEs in Figure 2.**
> For the 9 time periods of size 111, NFE=111 and NFE=1 respectively represent the small step manner and large step manner. Specifically, for the small step manner of NFE=111, we conduct the denoising process 111 times from $x_s$ to get $x_t^s$ within each time period. For the large step manner of NFE=1, we conduct the denoising process only once to get $x_t^l$. Comparing $x_t^s$ and $x_t^l$, we can get one data point for each time period, and thus 9 data points for the whole MSE curve in the right part of Figure 2.
>
> **5: Conversion between noise and data predictions.**
> The noise prediction and data prediction can be mutually converted through eq 1, $x_t = \alpha_t x_0 + \sigma_t \epsilon$. For input $x_t$, we can predict noise $\epsilon$ in noise prediction mode. Equally, we can derive the estimation of $x_0$ through the above equation.

---

> > ### Comment · Reviewer_heMq · 2024-11-25
> >
> > Thank you for addressing my concerns. Based on this and referring to the comments of other reviewers, I believe my original score still holds.

---

> > > ### Author Response · Authors · 2024-11-25
> > >
> > > Thanks again for your meticulous reviews. We are open for any potential issues.

---

### Official Review · Reviewer_P7sa · 2024-10-31

**Soundness:** 3
**Presentation:** 3
**Contribution:** 2
**Rating:** 3
**Confidence:** 3

**Summary:**

The paper proposes a modification to the sampling procedure in diffusion models to further reduce discretization error beyond that achieved with high-order solvers. The authors re-examine the sampling error at each time step and observe that the discretization error decreases over time. Building on this observation, they propose leveraging the denoiser at larger time steps to substitute for those at current steps, leading to reported improvements in performance.

**Strengths:**

The paper’s idea is interesting, and the presentation is clear and easy to follow. Addressing discretization error is a important topic, and this work proposes a novel approach in that direction.

**Weaknesses:**

Several aspects need further clarification:
-  **Main Observation on Discretization Error:** The central observation, illustrated in Figure 2, is that discretization error decreases over time. The y-axis is labeled with MSE,  could the authors clarify what quantity the MSE is measuring and how it directly relates to the discretization error?

- **Interpretation of Figure 2:**  The paper infers from Figure 2 that "the noise pattern is more recognizable as t increases,". But I feel there are other possible interpretations. For example, wouldn't it be possible that as sampling progresses from time T to 1, the discretization error accumulates, leading to an increase in error as we go backward in time?  I suggest the authors conduct an ablation study or provide theoretical justification to rule out this possibility.

- **Algorithm Procedure:** In equation (8), both terms on the right-hand side involve ϵθ ​ (with xθ also defined through ϵθ​). It’s unclear why only the ϵθ​ in the second term is evaluated at τ instead of t. Would it be beneficial if both terms were modified similarly? It would be valuable to discuss whether the authors explored modifying both terms and what the results were if so.

- **Similarity to Existing Methods:** In my opinion, the formulation of D_{t−1}^ours​ bears a strong similarity to equation (3) of [1].  Could the authors elaborate on the differences between the current method and that in [1] and provide a numerical comparison?

**Minor Issues:**
- The paper could benefit from enhanced self-containment. For instance, xθ is used before it is defined, and λt​ is not explicitly defined anywhere in the paper.
- The numerical results show a very mild improvement over baseline results. In Figures 4 and 5, the proposed method, on average, reduces NFE by no more than one for the same error level.
- Minor typos: Line 271, "general from" should be "general form"; Line 284, "between equation (9) and equation (9)" needs correction.


[1] Karras, Tero, et al. "Guiding a Diffusion Model with a Bad Version of Itself." arXiv preprint arXiv:2406.02507 (2024).

**Questions:**

Please see the weakness.

I would be open to raising the score if these concerns are adequately addressed.

---

> ### Author Response · Authors · 2024-11-22
>
> **1: Main observation on discretization error.**
> The MSE in Figure 2 reflects the distribution distance. In Section G of Appendix, we explain the rationality of MSE for distribution measurement in diffusion models. The MSE error between 1) the exact form manner and  2) the small step size manner, is caused by the existence of approximation error. The MSE error between 2) the small step size manner and  3) the large step size manner, is induced by the existence of discretization error.
>
> **2: Interpretation of Figure 2.**
> Thanks for the valuable suggestion. Your intuition of error accumulation itself is reasonable. However, for Figure 2, the MSE loss value is calculated within each small time period [$x_s, x_t$]. Starting from the exact form of $x_s$, the MSE value reflects the induced error at this time period, instead of from time T to 1. As for the error accumulation, it is also verified in Table 1 of the main manuscript. In this table, we calculate the MSE error from time T to 1, and find that our method has smaller total (accumulation) error.
>
> **3: Algorithm procedure.**
> We only modify the $\epsilon_\theta$ part in equation 8 with several reasons. First, equation 8 resembles the forward process $x_t = \alpha_t x_0 + \sigma_t \epsilon$, where $x_\theta$ estimates $x_0$ and $\epsilon_\theta$ estimates $\epsilon$. We thus only modify the $\epsilon_\theta$ part. Besides, this also elegantly results the $D_t$ of DDIM sampler from $\epsilon_\theta(x_t, t)$ to $(1+c) \epsilon_\theta(x_t, t) - \epsilon_\theta(x_\tau, \tau)$.
> From the performance perspective, we also find that this simpler implementation achieves better quality than modifying both $x_\theta$  and $\epsilon_\theta$ in equation 8.
>
> **4: Similarity to existing methods.**
> In work [1], $D_t$ denotes the denoising network. In our paper, as we emphasized in eq 6, $D_t$ is just a general notation in the solver formulation of eq 6. Indeed, we only modify $\epsilon_\theta(x_t, t)$ with $\epsilon_\theta(x_\tau, \tau)$, and this results in the formulation of eq 11 after derivation. This is the key difference. Besides, work [1] focuses on changing the guidance format to improve the generation quality. While, our method is designed for accelerating sampling from the perspective of reducing sampling error.
>
> [1] Karras, Tero, et al. "Guiding a Diffusion Model with a Bad Version of Itself." arXiv preprint arXiv:2406.02507 (2024).
>
> **5: Numerical results.**
> The FID metric may fail to fully reflect the improvements of our method. In Figure 10 and Figure 11 of the Appendix, we present the visual comparison of our method and the baseline sampler. Our method significantly reduces the sampling steps required to generate high-quality images. For example, our method can achieve comparable visual results with only 10 steps on DiT model compared to the baseline sampler with 50 steps
>
> **6: Improved expression and minor typos.**
> Thanks for the valuable feedback. $x_\theta$ represents data prediction, which can be mutually converted with noise prediction.  In Line 149-151, we briefly explain that time (t) domain can ber converted to log-SNR ($\lambda$) domain by the changeof-variables.  We will also add the meaning of $x_\theta$ and $\lambda_t$, and correct the typos in the revised version.

---

> ### Author Response · Authors · 2024-11-26
>
> Dear reviewer, we present detailed explanations in the last response to solve your confusion, and hope these address your concerns. We are looking forward to your reply.

---

> > ### Comment · Reviewer_P7sa · 2024-11-27
> >
> > Thank you for the response. However, I am still unclear about the precise computation of the MSE. In this paper, the MSE is used to quantify the distance between two distributions, which differs from its conventional application in measuring differences between images. Could the authors provide a clear mathematical formula for the MSE in the context of distributions? Additionally, it was mentioned that the MSE is computed over a small interval $[x_s, x_t]$. Could you elaborate on how $s$ is determined for each $t$? Furthermore, the explanation provided for why only the $\epsilon_\theta$ part is modified remains unconvincing. I share the concerns raised by another reviewer that the intuition behind the proposed method should align with its formulation. However, the current intuition does not seem to justify modifying only the $\epsilon_\theta$ part.

---

> > > ### Author Response · Authors · 2024-11-27
> > >
> > > Thank you very much for continued attention to our work and for the valuable feedback provided.
> > > Based on the feedback received during the rebuttal period, we are happy that it seems part of your concerns are addressed. Below we list the detailed explanations for your remained concerns and questions.
> > >
> > > **1. Precise computation of the MSE.**
> > > In Line 200-241 and caption of Figure 2, we detailly introduce the calculation of Figure 2 and MSE. Here, we give step-by-step explanation for better understanding.
> > > Specifically, within the time interval [s, t], we construct three distinct transition processes, each subjected to varying levels of sampling error. 1) We sample plenty of images from the training data, forming the pristine image distribution $P(x_0)$. 2) For the first transition process (exact form) in Figure 2, we can derive the data distributions $P(x_s)$ and $P(x_t)$, employing the transition kernel outlined in equation 1. The transition from distributions $P(x_s)$ to $P(x_t)$ is free of both approximation and discretization errors.  3) For the second transition process (small step size) in Figure 2, starting from the same $P(x_s)$, we iteratively employ the denoising network to get $P(x_t^s)$. The transition from $P(x_s)$ to $P(x_t^s)$ can be seen as free from discretization error due to small step size, and only has approximation error. 4) For the third transition process (large step size) in Figure 2, starting from the same $P(x_s)$, we employ the denoising network once with large step size to get $P(x_t^l)$. The transition from $P(x_s)$ to $P(x_t^l)$ suffers from both discretization and approximation errors due to large step size. 5) Thus, we calculate the MSE loss between process 1 and 2, which represents the approximation error. The MSE loss between process 2 and 3 presents the discretization error.
> > >
> > > About the rationality of adopting MSE to measure distribution distance. In Section G of Appendix, we explain the rationality of MSE for distribution measurement in diffusion models. Adopting MSE to measure the distributions divergence in diffusion models is grounded with both theoretical guarantee and sufficient empirical support from classical and representative papers. (1) Employing MSE to measure the distribution divergence in this special case is theoretically guaranteed. [1] discusses MSE as a special case of maximum likelihood estimation when the error follows a Gaussian distribution. [2] covers why MSE is a reasonable choice under the assumption of Gaussian noise. In the context of deep learning, [3] discusses the application of MSE, particularly in error measurement in generative models. Since in the context of diffusion models, gaussian distribution is the essential and default choice. MSE is thus a simple, reliable and rational metric to measure the distribution divergence, under the special case of gaussian distribution. (2) It is also a common practice of previous sampler papers, that employing MSE to measure distribution distance. For example, the main-stream samplers (our baselines), including DPM-Solver++ [4] and UniPC [5], also employs MSE (l2 distance) to compare the convergence error between the results of different methods and 1000-step DDIM, in the text-to-image model provided by stable-diffusion. Besides, [6] focuses on the discretization error and also proposes to leverage root mean square error (RMSE) to measure the distribution distance between one Euler iteration and a sequence of multiple smaller Euler iterations, representing the ground truth.
> > >
> > > [1] Box G E P, Tiao G C. Bayesian inference in statistical analysis[M]. John Wiley & Sons, 2011.
> > >
> > > [2] Murphy K P. Machine learning: a probabilistic perspective[M]. MIT press, 2012.
> > >
> > > [3] LeCun Y, Bengio Y, Hinton G. Deep learning[J]. nature, 2015, 521(7553): 436-444.
> > >
> > > [4] Lu C, Zhou Y, Bao F, et al. Dpm-solver++: Fast solver for guided sampling of diffusion probabilistic models[J]. arXiv preprint arXiv:2211.01095, 2022.
> > >
> > > [5] Zhao W, Bai L, Rao Y, et al. Unipc: A unified predictor-corrector framework for fast sampling of diffusion models[J]. Advances in Neural Information Processing Systems, 2024, 36.
> > >
> > > [6] Karras T, Aittala M, Aila T, et al. Elucidating the design space of diffusion-based generative models[J]. Advances in neural information processing systems, 2022, 35: 26565-26577.

---

> > > ### Author Response · Authors · 2024-11-27
> > >
> > > **2. The determination of s and t.**
> > > We divide the 1000 step into 9 time periods, each of size 111.  For example, for the first time period, the time range is [999,888], wherein s=999 and t=888.
> > >
> > > **3. Modifying only the $\epsilon_\theta$ part.**
> > > The intuition for modifying $\epsilon_\theta$ is simple and straightforward. Equation 8 is a combination of data ($x_\theta$) and noise ($\epsilon_\theta$), it is natural to only replace the latter noise ($\epsilon_\theta$) part. $x_\theta$ is viewed independently as the approximation of $x_0$. Some may hold that $x_\theta$ is implicitly correlated with $\epsilon_\theta$, via $x_\theta(x_t,t) = x_t/\alpha_t - \sigma_t/\alpha_t \epsilon_\theta(x_t,t)$. While, due to the existence of coefficient $ \sigma_t/\alpha_t $, slight modification of $ \epsilon_\theta(x_t,t)$ is amplified, significantly affecting the approximation of $x_0$.  Another reason for supporting this is that this reduces to equation 11 of $D_{t-1} = (1+c) \epsilon_\theta(x_t,t) – c \epsilon_\theta(x_\tau,\tau)$. This equation resembles the 2-order DPM-Solver sampler in equation 13, indicating the effectiveness of out method.

---

### Official Review · Reviewer_skPZ · 2024-11-04

**Soundness:** 3
**Presentation:** 2
**Contribution:** 2
**Rating:** 6
**Confidence:** 4

**Summary:**

The paper studied the sampling process for diffusion models. The paper categorized sampling errors into discretization error and approximation error, and pointed out that existing (high-order) solvers aim to reduce the discretization error. The paper on the other hand focused on the approximation error, and proposed a speedup framework to reduce the error. The method introduced a new noise estimation term in the sampling process, with a tune-able coefficient to reduce the sampling error. The method can be combined with existing samplers, and the effectiveness is demonstrated in various experiment settings.

**Strengths:**

1. The paper proposed an effective technique to improve samplers for diffusion models without introducing extra NFEs, especially in small NFE cases. The method can be readily combined with a wide range of existing samplers without post-training the diffusion models.

2. Extensive experiments were conducted to show the performance of the proposed method. Experiment settings vary across different condition types, sampler orders, guidance strategies, sampling space and datasets.

**Weaknesses:**

1. While the paper performed motivational experiments to explain discretization error and approximation error, the two types of error are not discriminated in a formal formulation. It is also not clear why the new noise estimation term in eq 10-12 can help reduce the approximation error. Here \tau is a variable time step and is set to T in experiments. The connection between the solver and the choice of \tau is not specified.

2. The writing of the paper may need improving. The notations in formal formulation of the sampling process can be more consistent. The citation format makes parts of the paper hard to read.

**Questions:**

1. Is there a principled way to set the values of c and \tau? Are they dependent on the order of solvers or specific solvers?

2. What is the performance of the proposed method for large NFEs? Is it able to improve the generation quality as NFE increases?

---

> ### Author Response · Authors · 2024-11-22
>
> **1: Formulation of the two errors and how to reduce approximation error.**
> In eq 7, we formulate the introduction of the discretization error and approximation error, where approximation error is induced by the network approximation of the score function and discretization error is caused by the Taylor expansion of the continuous integral. Eq 7 clearly discriminates and disentangles these two sampling errors.
> How to reduce approximation error? In Figure 2 and L260-269, we show that the approximation error decreases as the step t increases, and the network’s estimation is more desired at larger step. This motivates us to substitute the noise prediction at the current step t with that of a preceding larger step $\tau$. Thus, in eq 10, we replace the noise estimation part $\epsilon_\theta(x_t, t)$ with $\epsilon_\theta(x_\tau, \tau)$.
>
>
> **2: Writing and consistent notation.**
> Thanks for the valuable suggestions. We will improve the formulation consistency in the revised version.
>
>
> **3: Setting of c and $\tau$.**
> Through extensive experiments on these two hyperparameters in Figure 8 of the main manuscript, we find that linear strategy of c consistently achieves better performance than constant setting. For $\tau$, higher $\tau$ usually corresponds to better performance. These findings are also consistent with our analyses.
> Besides, we apply the generic values of c and $\tau$ determined under this principle to the samplers of various orders and find it consistently effective.
>
> **4: Performance on larger NFEs.**
> In Table 6 of Appendix D, we show the performance on larger NFEs. DualFast achieves comparable performance with NFE=10, compared to DDIM sampler with NFE=20. Generally, DDIM sampler can generate clear images with fine details under 20 NFE, and the image quality improvement is marginal when further increasing the sampling steps. Besides, we also show the visual comparisons on DiT in Figure 10 of the Appendix, where our method achieves comparable image quality with NFE= 10, compared to DDIM sampler with NFE=50.

---

### Meta-Review · Area_Chair_ft4D · 2024-12-20

**Metareview:**

The paper introduces a unified acceleration framework for Diffusion Probabilistic Models that addresses both discretization and approximation errors, enabling faster and more accurate sampling. Key contributions include a dual-error disentanglement strategy and compatibility with existing solvers.

The reviewers noted several positive aspects of the paper, but raised several doubts about the mathematical definitions and the method and accuracy of the calculations. These issues were not resolved through discussions between the authors and reviewers. Improvements will still be needed for this paper to be seen as sound.

**Additional Comments On Reviewer Discussion:**

skPZ is a bit more positive about the novelty of the paper. In contrast, P7sa and u2p2 responded that the definitions used in the results are unclear and the results cannot be interpreted. The authors attempted to refute this, but it remains unclear. heMq similarly raised concerns about the clarity of the results.

---

### Decision · Program_Chairs · 2025-01-22

Reject